



# Observed changes in the temperature dependence response of surface ozone under NOx reductions

Noelia Otero[1,2], Henning W. Rust[2], and Tim Butler[1,2]

[1]Institute for Advanced Sustainability Studies, Potsdam, Germany
[2]Institut für Meteorologie, Freie Universität Berlin, Germany

**Correspondence:** Noelia Otero (noelia.oterofelipe@iass-potsdam.de)

**Abstract.** Due to the strong temperature dependence of surface ozone concentrations ($O_3$), future warmer conditions may worsen ozone pollution levels despite continued efforts on emission controls of ozone precursors. Using long-term measurements of hourly $O_3$ concentrations co-located with $NO_x$ concentrations in stations distributed throughout Germany, we assess changes in the climate penalty, defined as the slope of ozone-temperature relationship during the period 1999-2018. We find a
stronger temperature sensitivity in the urban stations over the southwestern regions, especially in the first period of the study (1999-2008). We show a decrease in the climate penalty in most of stations during the second period of the study (2009-2018), with some exceptions (e.g. Berlin) where the climate penalty did not show significant changes. A key motivation of this study is to provide further insights into the impacts of $NO_x$ reductions in the $O_3$-temperature relationship. For that, we propose a statistical approach based on generalized additive models (GAMs) to describe ozone production rates, inferred from hourly
observations, as a function of $NO_x$ and temperature, among other variables relevant during the $O_3$ production. We find lower $O_3$ production rates during the second period (2009-2018) at most stations and a decreasing sensitivity to temperature, pointing out that lowering $NO_x$ concentrations resulted in decreasing $O_3$ production rates. However, we also observe changes in the shape of the function representing the $O_3$-temperature relationship, which indicate that $NO_x$ reductions alone can not explain the changes in the temperature dependence of $O_3$. Our analysis would suggest that decreasing $NO_x$ concentrations are not the
only factor causing the observed changes in the climate penalty factor.

## 1  Introduction

Tropospheric ozone ($O_3$) is a secondary pollutant formed from complex photochemical reactions of nitrogen oxides ($NO_x$), carbon monoxide (CO) and volatile organic compounds (VOCs) (J.H and Pandis, 2006). Changes in emissions of two of its major precursors, $NO_x$ and VOCs, might alter ozone formation regimes that are controlled by the initial $NO_x$/VOC ratio
(Sillman, 1999). Large $NO_x$ emissions and concentrations favour a VOC-sensitive regime, which is commonly found in urban areas, while large VOC emissions and concentrations, and $HO_x$ production rates favour a $NO_x$-sensitive regime, usually observed in rural environments (Sillman, 1999). The chemistry of $O_3$ production vary nonlinearly with temperature, which speeds up the rate of many chemical reactions. Furthermore, temperature is a fundamental variable that controls variations of biogenic emission of VOCs that increase with temperature and solar radiation (Pusede et al., 2015). Therefore, $O_3$ production





is highly sensitive to meteorological parameters, and thus, changes in ambient conditions and precursor emissions are nonlinear and complex.

A wide number of studies have shown that the $O_3$-temperature relationship varies in space and time due to differing chemi-
cal and meteorological mechanisms that influence $O_3$ formation (Rasmussen et al., 2013; Bloomer et al., 2010; Steiner et al., 2010). It has been recognized the temperature dependence of biogenic VOC emissions as well as the sensitivity of $O_3$ production to temperature to the peroxy acyl nitrate (PAN) dissociation rates (Jacob et al., 1993; Sillman and Samsom, 1995; Jacob and Winner, 2009). Moreover, dry deposition (Wesely, 1989) and $NO_x$ emissions (Coates et al., 2016) can contribute to the $O_3$-temperature relationship. Pusede et al. (2015) provides a comprehensive review of the temperature dependence of $O_3$
production. They pointed out that changes in $O_3$ precursors under a warmer climate will affect $O_3$ production in a predictable but complex way. For example, the continued $NO_x$ reductions in urban areas would lead to a transition in the chemistry of $O_3$ production into chemical regimes typically observed in rural areas.

Romer et al. (2018) investigated the effect of temperature in $O_3$ production using measurements in a rural site over south-eastern U.S. They found that local chemistry were key drivers of increased $O_3$ concentrations on hotter days, and a large
proportion of this increase was attributable to temperature-driven increases in soil emissions of $NO_x$. Recent modelling studies have examined the processes driving the $O_3$-temperature relationship. Porter and Heald (2019) used model simulations to quantify the contribution of mechanisms driving the $O_3$-temperature relationship. They found that a large proportion of the $O_3$-temperature relationship might be explained by other meteorological phenomena such as stagnation and humidity over Europe. Stagnant conditions characterised by low wind speed, allow $O_3$ to build up to high levels. High levels of humidity have
certain scavenging effect on $O_3$, as higher humidity is usually associated to greater cloud cover and atmospheric instability that can inhibit phochemical reactions and hence, decrease $O_3$. Similarly, Kerr et al. (2019) performed sensitivity simulations to examine the role of the processes related the to the $O_3$-temperature relationship over U.S., focusing on transport, chemistry and anthropogenic emissions. They found that atmospheric transport played a significant role in explaining the $O_3$-temperature relationship through out much of U.S. Since transport is indirectly related to temperature, the authors highlighted the importance
of providing a better understanding of the changes in the mechanisms linking transport and $O_3$ in a warmer climate.

Under future climate conditions, the benefits from control strategies of ozone precursors might be countered by temperature increases (Rasmussen et al., 2013). This effect has been termed in the literature as a "climate penalty", which has been used to quantify the additional increase of $O_3$ or the reduced benefits of emissions controls as a result of climate change (Wu et al., 2008; Rasmussen et al., 2013). Observational and modelling studies (Bloomer et al., 2009; Steiner et al., 2010; Rasmussen
et al., 2013) have reported a decreasing sensitivity of $O_3$ to temperature over time reflecting the emission reductions. Using observational datasets, Jing et al. (2017) examined the climate penalty factor during three periods covering a long period of 1990-2015. They found that the climate penalty, defined as the slope of $O_3$ change with increasing temperature, was by average $0.47\,\mathrm{ppb\,K^{-1}}$ less in the second part of the period (1999-2007) than in the first part (1990-1998), but this decrease was not observed in the last part of the period (2008-2015), despite the $NO_x$ reductions in most of the urban areas of Midwest U.S. Recently, Boleti et al. (2020) showed a decreasing sensitivity of $O_3$ to temperature during the period 2000-2015 over some





European regions. They suggested that a weaker $O_3$ sensitivity could be attributed to decreasing $NO_x$ concentrations and the differences in the changes to this sensitivity across sites were driven by regional meteorological conditions.

According to the EuroDelta-Trends modelling experiment (ETC/ACM, Colette et al., 2017) the reduction of European anthropogenic emissions of $O_3$ precursors was the main factor in decreasing summertime $O_3$ peaks episodes during the period 1990-2010. Several studies have documented downward trends across European sites of different metrics of ozone concentrations, such as the 4th highest daily maximum 8-hour ozone (4MDA8) and the number of days maximum maximum 8-hour ozone > 70 ppb (NDGT70) (Fleming et al., 2018; Chang et al., 2017). Some studies have reported increasing levels of $O_3$ concentrations at urban polluted sites as a result of lower titration processes through reaction with ambient nitric oxide (Yan et al., 2019; Querol et al., 2016). Based on measurements and sensitivity analysis Yan et al. (2018) showed that emission reductions had contrasting effects on $O_3$ and its interannual variability was regulated by climate variability. As stated in early studies, the current regulations on emissions of $O_3$ precursors, in particular $NO_x$, establish an ideal scenario for investigating the impacts of these changes on the $O_3$-temperature relationship (Pusede et al., 2015). This is crucial to improve air quality regulations because mechanisms influencing the relationship between $O_3$ and temperature are not completely well understood, partly because it is also influenced by meteorological processes and large-scale atmospheric patterns associated with high temperatures that lead to high $O_3$ concentrations.

Our work examines long-term $O_3$ concentrations to investigate the observed trend in the summertime climate penalty factor. We focus on Germany where the temporal homogeneity and diversity of the data offer an unique opportunity for long-term analysis of $O_3$ and $NO_x$. We examine changes in the $O_3$-temperature relationship over a 20-year time period covered 1999 to 2018, for which a greater number of sites were available. Furthermore, we restrict our study to summertime when $O_3$ normally reaches the highest levels and the photochemical activity is higher (Pusede et al., 2015). In addition, it has been shown a stronger temperature dependence of $O_3$ over Germany in summertime (Otero et al., 2018). We begin our study by calculating the trends in $NO_x$ concentrations that might lead to changes in the $O_3$-temperature relationship and then, we examine the climate penalty factor over the last two decades. Since the variability of $O_3$ production can explain a considerable proportion of $O_3$-temperature relationship, we propose an observational-based modelling approach to examine the nonlinear dependence of $O_3$ production on $NO_x$-temperature relationship. Within a statistical modelling framework built upon Generalized Additive Models (GAMs), we infer $O_3$ production (as a rate of change of $O_3$, $\Delta O_3$) from hourly $O_3$ concentrations. Thus, we model $\Delta O_3$ as a function of temperature and $NO_x$ along with other critical variables during the $O_3$ formation. Ultimately, we aim to provide new insights into the $O_3$ response to changes of its precursors in different environments and the effectiveness of emission reductions.

## 2 Data

Hourly measurements of $O_3$ and $NO_x$ concentrations were extracted from the European Environment Agency's (EEA) public air quality database "AirBase" (https://www.eea.europa.eu/data-and-maps/data/aqereporting-8). The number of sites and length of the period covered by each station for which measurements are available vary spatially and greatly by pollutant. The selection





of the monitoring stations with co-located data ($O_3$ and $NO_x$) was based on the station type (background), station type area (rural, urban, suburban) and altitude (<1000m). Only the stations reporting more than 75 % of valid data out of all the possible data in each summertime were included in the study. We use the stations with at least 19 years with hourly co-located data

within the whole period of study defined from 1999-2018. Here, summertime is referred to July-August-September (JAS), with a strong $O_3$-temperature relationship, particularly in Central Europe (Otero et al., 2018). A total of 29 stations meet the pre-processing criteria: 15 rural, 12 urban and 2 suburban stations. Despite that the spatial distribution of the measurement sites is not uniform with the largest density of stations over west and central Germany, a representative number of stations covering eastern regions are included (figure 1). Daily means and daily maximum of the running 8-hour mean of $O_3$ (MDA8)

were calculated following the European Union Directive of 2008 procedure (European Parliament and Council of the European Union, 2008).

The meteorology was extracted from the ERA5 (Herbach and Dee, 2016), the latest climate reanalysis produced by the European Centre for Medium Range Weather Forecast (ECMWF) that provides hourly data on regular latitude-longitude 0.25°x0.25° spatial resolution. The variables included in the analysis are air surface 2m-temperature (°C), 10m u and v-component of wind

($\mathrm{m\,s^{-1}}$), boundary layer height (m) and relative humidity at 1000 hPa (%).

## 3   Methods

### 3.1   Climate penalty factor

A number of definitions have been used in the literature to characterise the ozone climate penalty, usually represented as the linear relationship between $O_3$ and temperature. Climate penalty values are normally computed using daily maximum

summertime $O_3$ observations (1 or 8h average) and daily maximum temperature, although there is no standard definition (Pusede et al., 2015). Here, we adopted one of the most common metric to represent the climate penalty (hereinafter, $m_{O_3T}$) as the slope of the best fit line between long-term MDA8 concentrations and daily maximum temperature (Bloomer et al., 2009; Steiner et al., 2010; Otero et al., 2018). We first calculated the $m_{O_3T}$ with a linear regression model applied separately for each station and each period (1999-2008, 2009-2018). The general equation for the linear model can be written as follows:

$$Y(t) = N(\mu(t), \sigma^2) \tag{1}$$

Then, we examined the difference between the slopes obtained for each period, introducing an interaction factor term in the linear model to quantify the slope differences:

$$\mu(t) = T(t) * P(t) \tag{2}$$

Y(t), T(t) are the time series of MDA8 and daily maximum temperature (respectively) for the whole period 1999-2018 and P

is categorical variable with two categories: one representing the period 1999-2008 and another representing 2009-2018.



### 3.2 Approximation of $O_3$ production rates from observations

Most of the previous works have used numerical models (Steiner et al., 2006), box model (Coates et al. 2016), plume model (LaFranchi et al., 2011) or analytical models (Pusede et al., 2014; Romer et al., 2018) to analyse the temperature-dependent mechanisms affecting the $O_3$ production. Here, we propose a new approach based on GAMs to examine changes in the $O_3$ production. We approximate the later by the rate of change of hourly $O_3$ concentrations as:

$$\Delta O_3(t) = O_3(t) - O_3(t-1) \tag{3}$$

The general $O_3$ budget equation can be expressed as:

$$dO_3/dt = PO_{3chem} + LO_{3chem} + MD \tag{4}$$

$PO_{3chem}$ represents the chemical $O_3$ production rate, $LO_{3chem}$ is the chemical loss rate and the last term MD represents the dynamical processes that influence $O_3$ concentrations, including mixing and dry deposition processes. These individual processes can vary in strength and by location throughout the day.

As we aim to assess how $NO_x$ reductions influence the sensitivity of $O_3$ to temperature, we restrict our analysis to a time interval with an intense photochemical activity, which usually coincides with higher $O_3$ concentrations and warmer temperatures. Thus, the data was filtered to avoid including non-related photochemical processes that might mask the photochemistry in the daily $O_3$ production. First, we selected data after sunrise and until $O_3$ reaches the daily maximum value (usually in the afternoon). In order to exclude some maximum values that might occur late in the afternoon or evening and are mostly related to prevailing meteorological conditions and transport processes (Kulkarni et al., 1993), the time was restricted to 17:00 H (local time). Then, a wind speed condition was used to exclude the hourly data when wind speed was higher than $3.2\,\mathrm{ms^{-1}}$, which is the threshold value usually applied to define stagnant conditions (Horton et al., 2014). After a first inspection of the data, we found considerable differences in the minimum of $NO_x$ concentrations across some stations and periods, likely due to the detection methods. To better establish a comparison between stations and periods, we applied an additional filter to remove $NO_x$ values below $5\,\mathrm{\mu gm^{-3}}$. The number of observations that met these conditions varies with each station type and on average a 20% (urban), 14%(rural), 18%(suburban) of the total data was used.

### 3.3 Modeling $O_3$ production rates with GAMs

GAMs (Hastie and Tibshinari, 1990; Wood, 2006) were used to examine variations in $\Delta O_3(t)$ over the last two decades and the changes in the relationship $NO_x$-temperature given the observed downward trends of the $O_3$ sensitivity to temperature in the two periods of study 1999-2008 and 2009-2018. GAMs are useful tools for estimating non-parametric relationships whilst retaining clarity of interpretation (Wood, 2006). The relationship between the explanatory variables (henceforth covariates) and the response is described by smooth curves (splines, or potentially other smoothers). Such models have proven useful for studying the complex non-linear relationships between atmospheric chemical species and meteorological parameters (Carslaw et al., 2007; Jackson et al., 2009; Barmpadimos et al., 2011; Boleti et al., 2019). GAMs allow including nonlinear interactions





between covariates with different smoothers assumed for each covariate. The GAM can be formally written as:

$$g(\mu) = \beta_0 + f_1(X_1) + f_2(X_1) + \ldots + \epsilon \tag{5}$$

where $g$ is the link function, $X_n$ are the explanatory variables and $f_n$ are the non-parametric smoothing functions. $\beta_0$ is the intercept and $\epsilon$ is an error term. If the response can be assumed to be normally distributed, the canonical link function is the identity. After a closer inspection of the residuals at the individual sites, we found non normally distributed residuals with problems in the tails. Alternatively, we used a scaled $t$ distribution recommended for heavy tailed response variables (Wood

et al., 2016). Thin plate regression was used as smoothers to describe a nonlinear relationship between the response and 2 covariates (interaction) (Wood, 2006). The smoothness of each function is controlled by the number of knots or effective number of degrees of freedom. Here, the smoothing parameters were estimated by restricted maximum likelihood (REML) (Wood, 2006).

The challenge in building a model that captures a lage proportion of the variability of $\Delta O_3$ is to select the key covariates out

of a large number of potential variables. As stated in the previous section, changes in $O_3$ concentrations depend on local production, involving many chemical reactions that vary with temperature, loss mechanisms that are sensitive to meteorological conditions and transport processes. Therefore, we chose the variables that are expected to have a major influence on $O_3$ production (e.g. $NO_x$). The photochemical nature of $O_3$ production is strongly influenced by temperature. In particular, temperature increases biogenic emissions of VOCs, such as isoprene, from vegetation (Coates et al., 2016; Pusede et al., 2014). Thus, we

use temperature as a surrogate to represent changes in VOC, since biogenic VOC are emitted as an exponential function of temperature (LaFranchi et al., 2011; Pusede et al., 2014).

Daytime variation in the boundary layer height (BLH) significantly contributes to changes in $O_3$ production rates that tend to increase with a deepening BLH during sunny and warm days. (Haman et al., 2014). In addition to chemical and mixing processes, changes in $O_3$ concentrations are influenced by deposition. Therefore, additional covariates are the percentage of

change of the boundary layer height growth rate ($\Delta BLH$) (in %) accounting for mixing processes, and vapour pressure deficit (VPD) as it has been recognised as a key variable for dry deposition (Kavassalis and Murphy, 2017; Otero et al., 2018). The VPD was calculated from the corresponding hourly data of air temperature and relative humidity. Moreover, we included the $O_3$ concentrations from the previous hour ($C_{O_3}(t-1)$) and the MDA8 concentrations from the previous day ($C_{MDA8}(t-24)$) to represent the persistence of previous chemical conditions, (Pusede et al., 2015). We first started with a baseline model that

included the nonlinear relationship between $NO_x$ and temperature as follows:

$$\Delta O_3 \sim f(T, NO_x) \tag{6}$$

$f(T, NO_x)$ represents the interaction between temperature (T) and $NO_x$ concentrations and it is included as a tensor product (Wood, 2017). Observing the skewness of the $NO_x$ data led us to introduce a modification in the baseline model using a log transformation of $NO_x$. Since we aim to build a parsimonious model to better explain the variability of $\Delta O_3$, we gradually

added the covariates to the baseline model through a selection procedure. During the stepwise process, we also allowed interactions between two influencing covariates: the VPD and the $C_{O_3}(t-1)$, and the $\Delta BLH$ and $C_{MDA8}(t-24)$. As in previous





studies (e.g. Gong et al., 2017), we adopt a forward selection method based on the Akaike information criterion (AIC) with the goal of obtaining a common model well defined across all of the stations. AIC is a robust approach to assess the model performance and to comparing the different model structures (Pedersen et al., 2019). The model selection procedure was applied separately at each station and period. Thus, we fit a GAM for the first period 1999-2008 (GAM-P1) and a GAM for the

second period 2009-2018 (GAM-P2). The model performance was assessed through standard diagnostic plots: QQ plots of the deviance residuals, scatter plots of the residuals against the fitted values, histogram of residuals and scatter plots the response against the fitted values (Wood, 2006).

## 4 Results

### 4.1 Climate penalty

Before calculating the $m_{O_3T}$, we assess changes in the $NO_x$ concentrations over the whole period of study (1999-2018). For that, we examine time series of the annual 5th, 50th, and 95th percentiles calculated from daily $NO_x$ concentrations, assessing the trends (Kendall, 1975) and estimating its slope (Theil, 1950; Sen, 1968). Figure 2 shows annual 5th, 50th, and 95th percentiles calculated from daily $NO_x$ concentrations at some example stations located in Berlin, Rhineland-Palatinate and Saxony that are representative for each station type area and will be used below to present the modelling results. The

$NO_x$ concentrations at the 95th percentile have generally declined over the overall period of study (1999-2018), but the most dramatic reduction is observed during the first part ot the period (1999-2008) in the example stations. Larger decreases are observed at the stations in Rhineland-Palatinate, specially at the urban station (DERP025) where the $NO_x$ concentrations at the 95th percentile declined at the rate of $-4.45\,\mu\,\mathrm{g\,m^{-3}yr^{-1}}$ in the first period 1999-2008 and $-3.38\,\mu\,\mathrm{g\,m^{-3}yr^{-1}}$ in the second period 2009-2018 (see Fig. S1 in the Supplementary Material). Similar trends are observed at the urban stations located in

the southwest and central regions (Fig. S1 and S2). The $NO_x$ concentrations at the 95th percentile have been reduced at the urban and rural stations in Berlin during the first period 1999-2008 with decreasing rates of -2.78 and $-1.77\,\mu\,\mathrm{g\,m^{-3}yr^{-1}}$, respectively, while small and non significant changes are observed during the second period (Fig. S1). Overall, annual 50th percentile $NO_x$ concentrations show a steady decrease in most of the stations of the study, more pronounced during the first period, and small changes are found at the 5th percentile of $NO_x$ especially during the second period 2009-2018 (Fig. S1 and

S2).

Figure 3 shows the spatial distribution of the $m_{O_3T}$ for each period. The highest values are found in the southwest stations during the first period 1999-2008 with $m_{O_3T}$ 5-6.5 $\mu\,\mathrm{g\,m^{-3}°C^{-1}}$. Among these sites, urban stations show a higher sensitivity to temperature compared to suburban and rural stations. The lowest values of $m_{O_3T}$ during the first period are observed in the north and eastern stations (4-5 $\mu\,\mathrm{g\,m^{-3}°C^{-1}}$). Significant differences between the $m_{O_3T}$ calculated for each period are observed in

most of the stations, including some rural areas in the southern regions where the $m_{O_3T}$ dropped $-1.2\,\mu\,\mathrm{g\,m^{-3}°C^{-1}}$ (Fig. S3). Only a few stations show similar values of $m_{O_3T}$ in both periods (e.g. Berlin). Boleti et al. (2020) reported a general decreasing sensitivity of daily maximum of $O_3$ with temperature for a shorter period (2000-2015) in regional clusters defined over Europe. They found larger trends in $m_{O_3T}$ at high and moderate polluted clusters ant they argued that it might be due to $NO_x$ reductions.





Here, we found a general decrease in $m_{O_3T}$ obtained from long-term data across different environments (i.e. rural, urban and suburban). Our results also pointed out significant differences in the $m_{O_3T}$ across stations, with some polluted areas where the $m_{O_3T}$ did not show significant changes with time (e.g. Berlin). As stated in the introduction, mechanisms controlling $m_{O_3T}$ are not well established. A priori it is not evident what the impact of $NO_x$ reductions is in the $O_3$ sensitivity to temperature, in particular in rural environments. Therefore, we next examine the variability of $\Delta O_3$ as a function of temperature and $NO_x$ in

order to provide further insights into the nonlinear temperature-dependence of $NO_x$ and the potential impacts on the observed $m_{O_3T}$.

## 4.2   Model performance

A final model including three interaction terms was designed from the selection procedure as the best fit to capture the $\Delta O_3$ variability at most stations and periods. The model selection indicated that as variables were added and the model complexity

increased (i.e. more interactions), the AIC decreased and the deviance explained increased (Fig. S4). The performance of the GAMs was assessed by the adjusted r-squared for the model ($R^2$), defined as the proportion of the variance explained (Fig. 4). The results showed similar $R^2$ values in both periods over most of the stations, with some exceptions where GAM-P1 seem to perform better than GAMP-P2 (e.g. over the region of Hessen). In general, GAMs showed a better performance over urban and suburban stations and 40% of the $\Delta O_3$ variability was captured. The models performed poorly when applied to rural stations,

they showed lower values of $R^2$. This likely reflects that GAMs designed with the underlying assumptions of the interactions between the selected covariates is better suited for urban and suburban areas than for rural regions.

## 4.3   Model interactions

Our approach is built upon a conceptual model (4) to evaluate the effect of chemical, deposition and dynamical-mixing processes affecting the $O_3$ production. The final GAM includes three interaction terms defined by the covariates T-$NO_x$, VPD-

$C_{O_3}(t-1)$ and $\Delta BLH$-$C_{MDA8}(t-24)$. Given that $\Delta O_3$ is modelled with GAMs separately at each station and period, a large number of interaction surfaces were obtained. Thus, we focus on a representative number of stations for each station type area (i.e. rural, urban and suburban). The example stations presented here were selected based on a relatively good performance of the model as well as the corresponding geographical location in order to examine the results from the previous section showing marked differences in the sensitivity of MDA8 to temperature. Figures showing the results obtained for the rest of stations

are available in the Supplementary Material. Note that the contour plots presented below reflect the partial effects, which allow us to compare the effect of those covariates included in the interaction term without considering the intercept and the other covariates (e.g. Fig. 5). The summed effects that include the intercept and constant values for the others covariates not shown in the interaction surface, presented similar shapes but with the additive effect of those constant values (not shown). To estimate the predicted surfaces within a range of data sufficiently supported by the observations, we used the first and the third quantile

of the distribution of the corresponding covariates for each station type area (urban, rural and suburban).



### 4.3.1 NO$_x$ and temperature

Figure 5 shows $\Delta O_3$ as a function of NO$_x$ concentrations and temperature for the example urban stations located in Berlin (DEBE034) and in Rhineland-Palatinate (DERP025) (see Fig. 1). Also shown in Fig. 5 are the estimated regression lines for temperature while holding constant NO$_x$ concentrations (i.e. mean conditions of NO$_x$ each period). As we aim to assess the impact of NO$_x$ reductions in the O$_3$-temperature relationship, we also use the GAM-P1 to project the $\Delta O_3$ response to temperature, as it has been estimated under the first period conditions, but using the mean NO$_x$ concentrations of the second period

2009-2018. Examining the GAM-P1 projection for the second period 2009-2018 and the GAM-P2 estimations can provide useful insight into the changes in the $\Delta O_3$ sensitivity to temperature when lowering NO$_x$ concentrations.

The interaction surfaces obtained from both stations illustrate the temperature dependence of $\Delta O_3$ with increasing temperatures, implying a VOC-limited chemistry (Fig.5, left). The temperature dependence of $\Delta O_3$ is observed to vary with NO$_x$, but also with temperature in both stations. We found a stronger temperature dependence of $\Delta O_3$ in the first period 1999-2008

(GAM-P1) compared to second period 2009-2018 (GAM-P2). This feature is more pronounced in Rhineland-Palatinate, where the mean NO$_x$ conditions declined by 35% in the second period 2009-2018 (relative to the first period 1999-2018), while in Berlin NO$_x$ declined only by 7.5%.

    We examine the $\Delta O_3$ response to temperature under the mean NO$_x$ conditions for each period using GAM-P1 and GAM-P2 along with the prediction obtained from GAM-P1 that projects the $\Delta O_3$ response in the second period 2009-2018 (prediction

line in Fig. 5, rigth). In Berlin, the $\Delta O_3$ response to temperature shows a similar increase with temperature in both periods. In this case, the GAM-P1 prediction for the second period 2009-2018 is in a good agreement with the shape obtained from GAM-P2, which suggest that a decreasing temperature sensitivity of $\Delta O_3$ could be explained by NO$_x$ reductions. The increase of $\Delta O_3$ with temperature is also depicted in Rhineland-Palatinate. The prediction from GAM-P1 for the second period 2009-2018 reveals discrepancies at higher temperatures when comparing to the $\Delta O_3$ response from GAM-P2. It can be noted that

the prediction from GAM-P1 for the second period (prediction line, Fig.5) does not capture the steepness at temperatures above 20°C showed by GAM-P2. Contrasting to the results in Berlin, the changes in the shape that represents the $\Delta O_3$ as a function of temperature suggest that the NO$_x$ reductions would only partially explain the observed changes in the O$_3$-temperature relationship, but rather an underlying effect is likely to influence the $\Delta O_3$ at higher temperatures. We found similar features in the rest of the urban stations than in the example stations, with consistent interaction surfaces in terms of the $\Delta O_3$ response to

NO$_x$ and the temperature dependence (Fig. S5). As in Rhineland-Palatinate, the regression lines were slightly different when comparing GAM-P2 and the projected $\Delta O_3$ response under NO$_x$ reductions (Fig. S6), which reinforce our hypothesis of an underlying factor influencing the $\Delta O_3$-temperature relationship.

    We further assess the effect of the temperature and NO$_x$ on $\Delta O_3$ separately with GAM-P1 and GAM-P2 under fixed NO$_x$ and temperature conditions determined as the 10th, 50th and 90th percentiles of the corresponding distributions over the whole

period of study (1999-2018). In contrast to the contour plots (Fig. 5), we now include the intercept and a constant value (i.e. median) for the rest of the covariates, in order to further examine the summed effects. Table 1 summarizes the values of the covariates for the selected stations. Figure 6 shows $\Delta O_3$ as a function of temperature. $\Delta O_3$ estimates are generally





lower in the second period 2009-2018 under moderate (50th) and high (90th) $NO_x$ concentrations at both stations. Similarly, Fig. 7 illustrates the changes in the nonlinear relationship between $\Delta O_3$ and $NO_x$. In general, at lower temperatures (10th)

$\Delta O_3$ decreases with increasing $NO_x$ concentrations. In Berlin, the relationships are similar for both periods, but show lower $\Delta O_3$ estimates in the second period. In Rhineland-Palatinate we found a steeper decreased of $\Delta O_3$ when moving to higher $NO_x$ concentrations during the second period. This indicate that the $\Delta O_3$ sensitivity to higher $NO_x$ at moderate and high temperatures is lower in the second period.

Figure 8 depicts the interaction surfaces for two selected rural stations located in the same regions than the urban stations

presented above, Berlin (DEBE032) and Rhineland-Palatinate (DERP017). Similarly than in the urban case, the temperature dependence of $\Delta O_3$ is stronger in the first period 1999-2008 compared to second period 2009-2018. The GAMs-P2 show a decreasing sensitivity of $\Delta O_3$ to temperature and $\Delta O_3$ is generally lower with increasing temperature under similar conditions of $NO_x$. We see similarities between the rural and urban stations in Berlin, in terms of the shape of the nonlinear relationship between temperature and $NO_x$, which is expected due to the proximity between both stations ( Fig. 1). In contrast to the urban

stations, the contours obtained from GAM-P1 and GAM-P2 are significantly different, particularly in Rhineland-Palatinate. The $\Delta O_3$ as a function of temperature under $NO_x$ mean conditions is also shown in Fig.8 (rigth). In Berlin, $NO_x$ concentrations declined by 28.8%, which could partially explain the decrease in $\Delta O_3$ estimates during the second period 2009-2018. However the shapes of the regression lines obtained from the GAM-P2 and the projected $\Delta O_3$ response from GAM-P1 differ at temperatures below 20°C (blue and green lines, Fig. 8). In Rhineland-Palatinate the temperature dependence is consider-

ably lower than in Berlin and a flat regression line is shown by GAM-P2 for the second period with a 37% decrease of $NO_x$ concentrations. The discrepancies between the projected $\Delta O_3$ response and the $\Delta O_3$ estimates from GAM-P2 are higher at temperatures below ~20°C. Overall, we found a larger variability among the rest of the rural stations considered in the study, in terms of the interaction surfaces $NO_x$-temperature (Fig. S7). This is also reflected in the estimated temperature response of $\Delta O_3$ when comparing GAM-P2 and the projected response using GAM-P1 (Fig.S8).

Figure 9 shows $\Delta O_3$ as a function of temperature under low (10th), medium (50th) and high (90th) levels of $NO_x$ at those rural stations. The differences between the periods are more evident in Rhineland-Palatinate where the regression line corresponding to the second period 2009-2018 becomes flat at temperatures between 18-22 °C at moderate (50th) and high (90th) $NO_x$ concentrations. In Berlin, $\Delta O_3$ slightly decreases in the second period, and the regression lines are very similar at the fixed $NO_x$ conditions in both periods. The variations of $\Delta O_3$ with $NO_x$ at different temperature conditions are shown in figure

10. While in Berlin the relationship is similar in GAM-P1 and GAM-2 for all temperature conditions, in Rhineland-Palatinate increases of $NO_x > 7\,\mu\,gm^{-3}$ indicate a major decreased of $\Delta O_3$ at medium (50th) and high (90th) temperatures in the second period.

Only two suburban stations were included in this study, in Berlin (DEBE051) and in Saxony (DESN045), both eastward located. The contours obtained in each period and station showed similar patterns than those found for urban stations, specially

in Berlin (Fig. S9). The GAMs consistently reproduce the temperature dependence of $\Delta O_3$ at higher temperatures and the differences between the GAM-P2 and the projected $\Delta O_3$ response to temperature with GAM-P1 were more evident in Saxony (Fig. S9, rigth).





### 4.3.2 VPD and O$_3$ from the previous hour ($C_{\mathrm{O_3}}(t-1)$)

Figure 11 reveals the nonlinear relationship between VPD and the $C_{\mathrm{O_3}}(t-1)$ at the selected urban stations in Berlin and in Rhineland-Palatinate. In general, $\Delta$O$_3$ tends to increase with higher levels of VPD (i.e. drier conditions) and low O$_3$ concentrations from the previous hour in both locations and periods. In the first period, the contribution of the interaction between VPD and persistent O$_3$ concentrations is similar at both locations, and the model shows maximum $\Delta$O$_3$ at $C_{\mathrm{O_3}}(t-1) < 30\,\mu\mathrm{gm}^{-3}$ and VPD $> 0.70\,\mathrm{kPa}$. In Berlin, the results obtained from GAM-P2 suggest that higher levels of VPD and low $C_{\mathrm{O_3}}(t-1)$ ($\sim 30\,\mu\mathrm{gm}^{-3}$) lead to an increase of $\Delta$O$_3$, but the $\Delta$O$_3$ tends to decrease faster with high $C_{\mathrm{O_3}}(t-1)$ concentrations (above $50\,\mu\mathrm{gm}^{-3}$) when comparing to GAM-P1. The interaction surfaces obtained in Rhineland-Palatinate show small changes when comparing both periods.

The contours obtained from the GAMs built for the rural stations are shown in Fig. 12. We see significant differences between these rural stations. $\Delta$O$_3$ dependence with VPD is more pronounced in Rhineland-Palatinate, especially in the second period 2009-2018 with a larger increase of $\Delta$O$_3$ with increasing VPD levels (i.e. drier conditions). In this case, GAM-P1 shows little changes in the estimated $\Delta$O$_3$ ($\sim 3\,\mu\mathrm{gm}^{-3}$) at low $C_{\mathrm{O_3}}(t-1)$ concentrations for all range of VPD, while the GAM-P2 shows a significant increase of $\Delta$O$_3$ under similar $C_{\mathrm{O_3}}(t-1)$ concentrations when moving to higher VPD. In Berlin, $C_{\mathrm{O_3}}(t-1)$ concentrations seems to have a major influence on $\Delta$O$_3$, and $\Delta$O$_3$ estimates are slightly lower in the second period 2009-2018 than in the first period 1999-2008. The interaction between VPD and $C_{\mathrm{O_3}}(t-1)$ in the suburban stations (Berlin and Saxony) is consistent with the patterns found in the urban and rural stations and $\Delta$O$_3$ increases with higher VPD and low $C_{\mathrm{O_3}}(t-1)$ concentrations (Fig. S10).

VPD is crucial and controls the stomatal conductance. Its effects can be summarised as follows: under high VPD levels (associated with high temperatures), plants cannot extract sufficient moisture from dry soils to satisfy the atmospheric demand for evapotranspiration (Teuling, 2018). In this situation of drought stress, plants close their stomata to reduce water loss and limit the uptake of ozone by vegetation.

Our results illustrate that the combination of high VPD and lower $C_{\mathrm{O_3}}(t-1)$ concentrations result in higher $\Delta$O$_3$ (thus, less uptake of O$_3$). Moreover, given that O$_3$ concentrations are typically lower in urban environments due to the local scavenge of O$_3$ (NO titration), a larger contribution of the interaction of VPD and $C_{\mathrm{O_3}}(t-1)$ to $\Delta$O$_3$ in the urban and suburban stations than in the rural stations is expected.

### 4.3.3 $\Delta BLH$ and MDA8 from the previous day ($C_{MDA8}(t-24)$)

The effect of mixing processes was introduced in the GAMs through the $\Delta BLH$ and $C_{MDA8}(t-24)$. Figures 13 and 14 depict the interaction surfaces between the covariates $\Delta BLH$ and $C_{MDA8}(t-24)$ at the selected urban and rural stations in Berlin and Rhineland-Palatinate. In general, $\Delta$O$_3$ is mainly dependent on changes in $\Delta BLH$ and it increases at higher $\Delta BLH$, while the influence of $C_{MDA8}(t-24)$ on $\Delta$O$_3$ is negligible for $\Delta BLH \sim < 30\%$. The results obtained from most of the stations at different environments (i.e. urban and rural) showed consistent shapes with the patterns described for the selected stations (not shown). Moreover, we found similar patterns for the suburban stations(Fig. S11).



These interaction surfaces can be used to interpret the nonlinear relationship between $\Delta BLH$ and $C_{MDA8}(t-24)$ concentrations. As BLH grows, air is entrained from layers aloft and $O_3$ production rates can increase or decrase depending on the $O_3$ concentrations in this residual layer (Haman et al., 2014). We show that a rapid development of the BLH along with high $C_{MDA8}(t-24)$ (from the previous day), likely stored at the residual layer, lead to an increase of $\Delta O_3$. Note that $C_{MDA8}(t-24)$ concentrations seems to have an influence on $\Delta O_3$ when the BLH rapidly changes. The effect of this interaction was slightly

larger in most of the urban and suburban stations as compared to the rural stations, while small differences are observed when comparing the patterns obtained from each period.

## 5    Summary and conclusions

We have examined the long-term $O_3$ sensitivity to temperature, as well as the modulation of this sensitivity by $NO_x$ concentrations, in a total of 29 stations over Germany during the period 1999-2018. Consistent with previous work, $O_3$ tends to increase

strongly with temperature under high $NO_x$ conditions due to increased in-situ photochemical production, while lower levels of $NO_x$ leads to a reduced $O_3$ sensitivity to temperature. Also consistent with previous work, we see a decreasing sensitivity of $O_3$ to temperature over our study period, coinciding with decreasing trends in NOx concentration.

In order to explain the trends in photochemical ozone production over our study period, we divided this period into two halves (1999-2008 and 2009-2018) and constructed sets of Generalized Additive Models (GAMs) based on hourly station

observations of ozone and $NO_x$ concentrations, along with temperature, vapor pressure deficit, and boundary layer height from a reanalysis product. We modeled the daily increase in $O_3$ concentrations as a function of three interaction terms accounting for phochemical production (dependent on $NO_x$ and temperature), dry deposition (dependent on vapor pressure defict and ozone concentrations from the previous hour) and mixing processes (dependent on the boundary layer height growth rate, and ozone concentrations from the previous day).

In most of the stations, the effect of the interaction term $NO_x$-temperature was larger in the first period than in the second period, resulting in higher $\Delta O_3$ estimates in 1999-2008 compared to $\Delta O_3$ estimates in 2009-2018. A decreasing sensitivity of $\Delta O_3$ to temperature was shown by the GAMs built for the second period 2009-2018 when comparing with the GAMs from the first period 1999-2008, leading to lower $\Delta O_3$ values under moderate-high temperatures in the second period. This decreasing temperature sensitivity was more pronounced in the southward urban stations (e.g. Rhineland-Palatinate). Thus, the lower $NO_x$

concentrations during the second period 2009-2018 resulted in a decrease in $\Delta O_3$ as well as a lower temperature dependence.

However, our results pointed out that $NO_x$ reductions can only partially explain the changes in the $O_3$-temperature relationship. Using the GAMs derived from the first period 1999-2008 to project the $\Delta O_3$ response to temperature under the mean $NO_x$ conditions of the second period 2009-2018, we showed that the shape of the regression lines have changed in the second period for a large number of urban stations. Similar conclusions were obtained for most of the rural stations, where the shape

of the projected $\Delta O_3$ response with temperature in the second period 2009-2018 differ from the estimated $\Delta O_3$ response from the GAMs built for that period.





We conclude that $NO_x$ reductions have had an influence in the decreasing temperature sensitivity of $O_3$, as shown in the GAMs for the second period 2009-2018, but that such reductions alone can not explain the changes in the observed $O_3$-temperature relationship. We interpret these discrepancies as an underlying effect influencing the $\Delta O_3$ that has not been
included in the model.

The temperature-dependence of bioginec VOC emissions is well-known. In particular, biogenic isoprene emissions have a strong temperature dependence with critical implications on $O_3$ production, mostly during warmer summer days. Therefore, one plausible explanation for the changes in the shapes of the $\Delta O_3$-temperature relationship might be attributed to the accompanying effect of changes in biogenic emissions (along with $NO_x$) that are likely influencing the temperature-dependence of
5  $\Delta O_3$ and consequently the $m_{O_3 T}$. We have shown a general decrease of $\Delta O_3$ at higher temperatures, which may suggest that enhanced temperature-driven biogenic emissions can result in $\Delta O_3$ being more dependent of $NO_x$ ($NO_x$-limited) and thus, more sensitive to the $NO_x$ controls. Our results have important implications for the implementation of mitigation strategies, specially when considering the effects of a warming climate. We expect that the methodology described herein can be applied to other locations with available long-term measurements to assess how $NO_x$ reductions have influenced the temperature dependence of $O_3$. Consistent with previous work, we may anticipate that our approach will show changes in the climate penalty factor as well as in the sensitivity of $\Delta O_3$ with temperature. Further analysis to examine in more detail the effect of $NO_x$ reductions in particular locations should be required.

5  In summary, the sensitivity of $O_3$ to temperature has decreased during the last period (2009-2018) over a great number of the German stations considered in the study, including rural areas. Even though $NO_x$ reductions accomplished during the last decades have partially counteracted the $O_3$ climate penalty, our study highlights the relevance of considering the influence of additional factors controlling the $O_3$-temperature relationship. Since observations of long-term dataset of VOCs are lacking, further analysis including short-term measurements of a suite of VOCs would be definitively required to quantify their contribution to the observed changes in the climate penalty.



# 6 List of figures



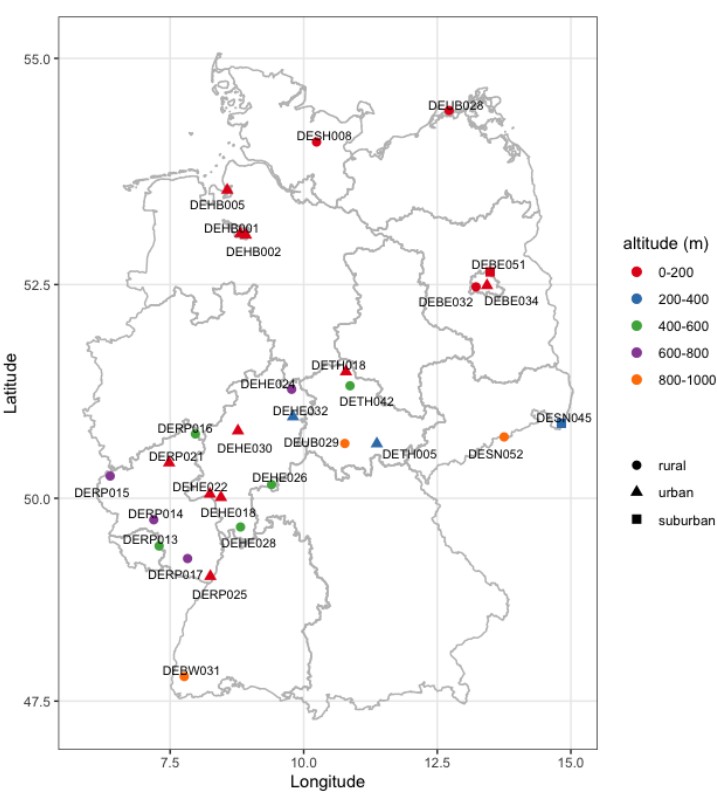

**Figure 1.** Spatial distribution of measurement stations. Network codes are indicated in text. Shapes indicate the station type area and color the altitude.



**Figure 2.** Time series of annual 5th, 50th, and 95th percentile of NOx concentrations for the whole period of study (1999-2018) at the example stations.





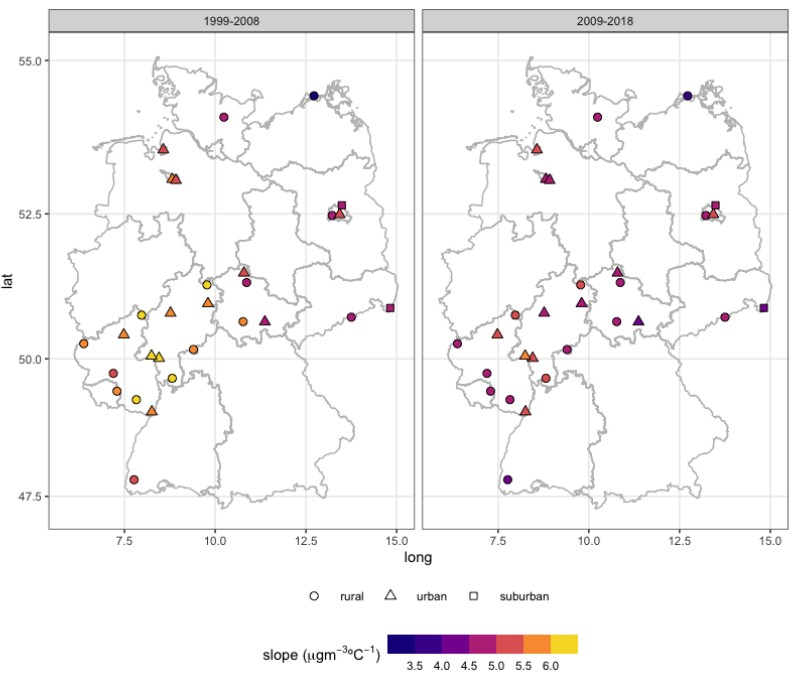

**Figure 3.** Spatial distribution of climate penalty factor calculated at each stations and period.

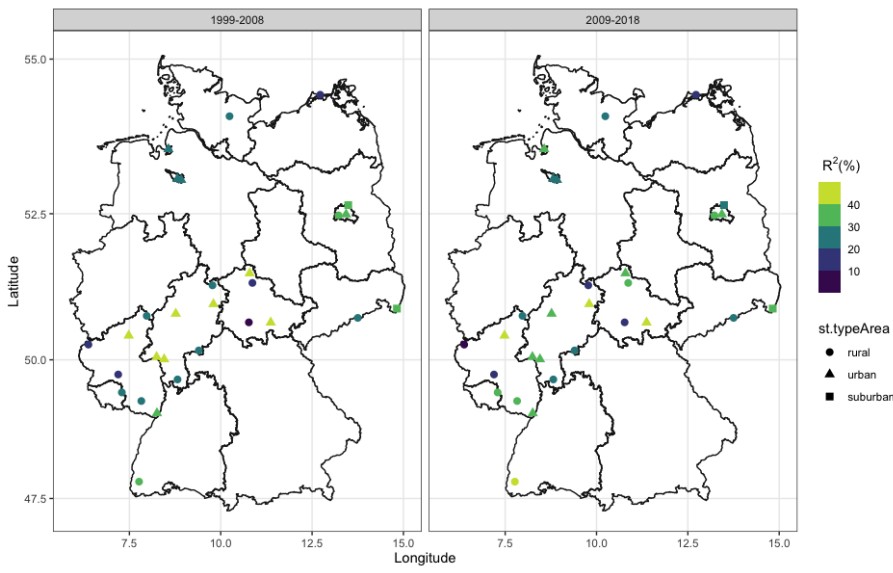

**Figure 4.** Spatial distribution of the adjusted r-squared for the model, $R^2$, for GAM-P1 (left) and GAM-P2(right).



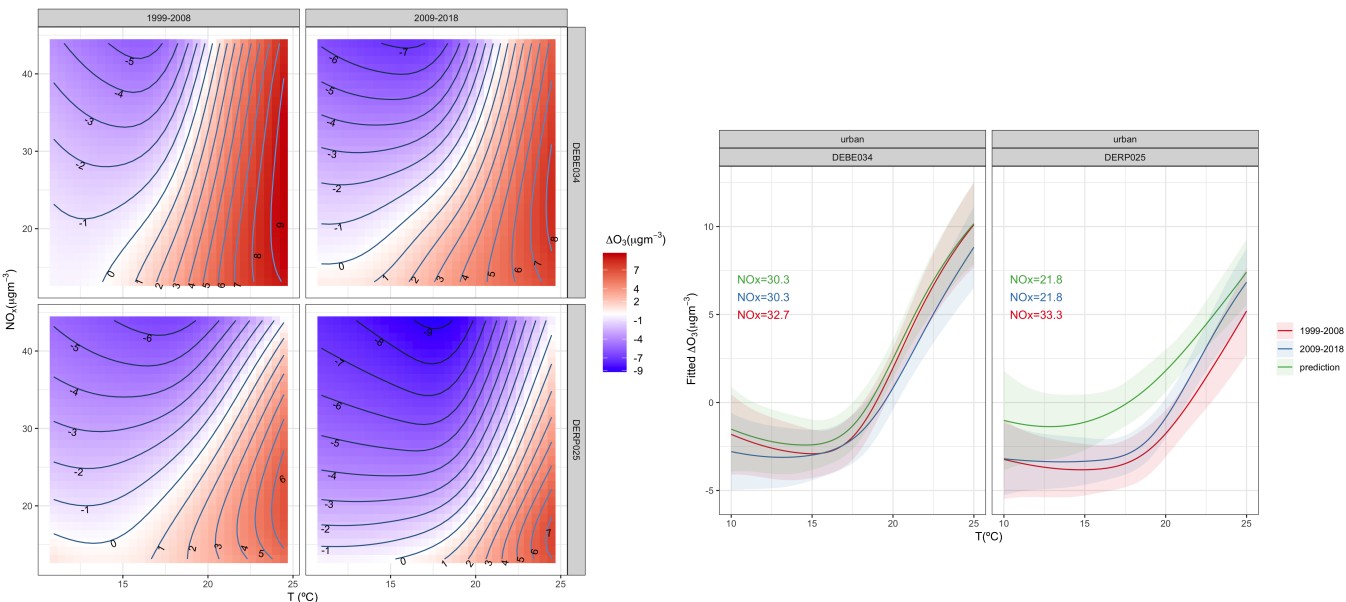

**Figure 5.** Contour plot for the interaction temperature-NO$_x$ at the urban stations in Berlin (DEBE034) and Rhineland-Palatinate (DERP025) for the first period 1999-2008 and second period 2009-2018 (left). In the right panel, smooth functions representing the temperature response of O$_3$ production rates under mean conditions of NOx (indicated by the text numbers) obtained from GAM-P1 (red line) and GAM-P2 (blue line), along with the prediction of the O$_3$ response using GAM-P1 (green line). Shaded bands represent the pointwise 95% confidence interval.




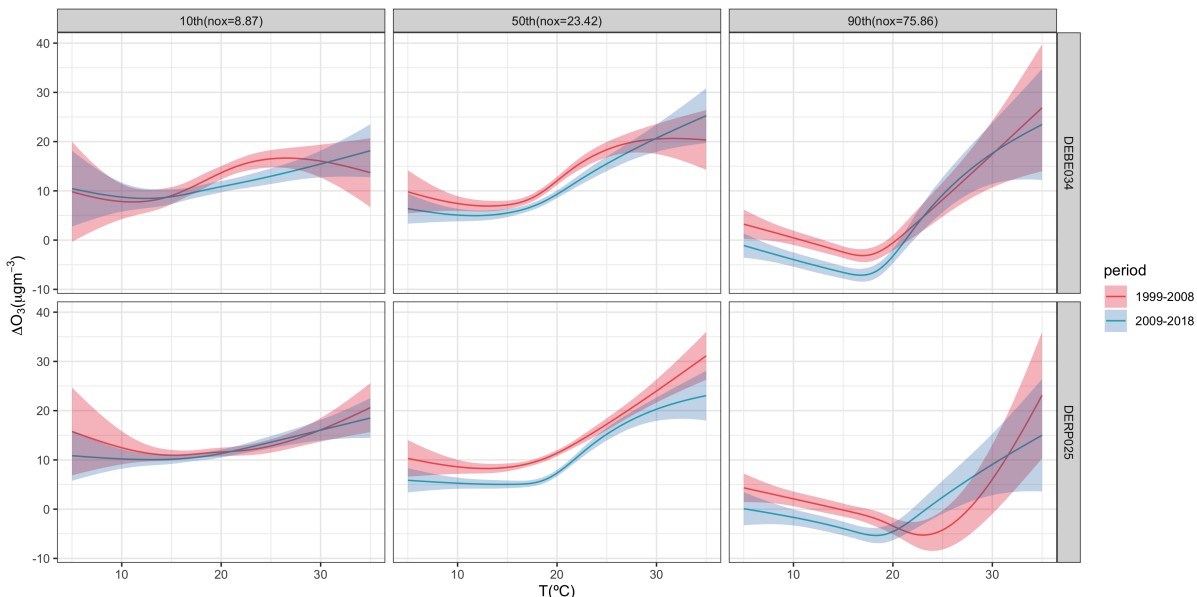

**Figure 6.** Smooth functions for temperature at low (10th), medium(50th) and high(90th) NO$_x$ conditions. Shaded bands represent the point-wise 95% confidence interval.

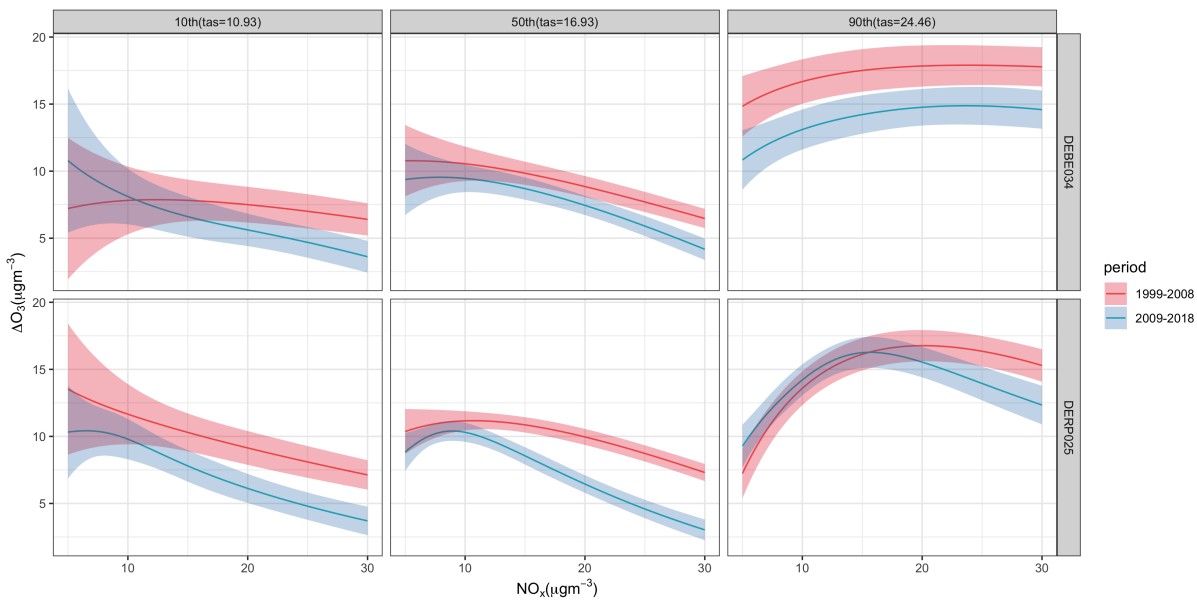

**Figure 7.** Smooth functions for NO$_x$ at low (10th), medium (50th) and high (90th) temperature conditions. Shaded bands represent the pointwise 95% confidence interval.





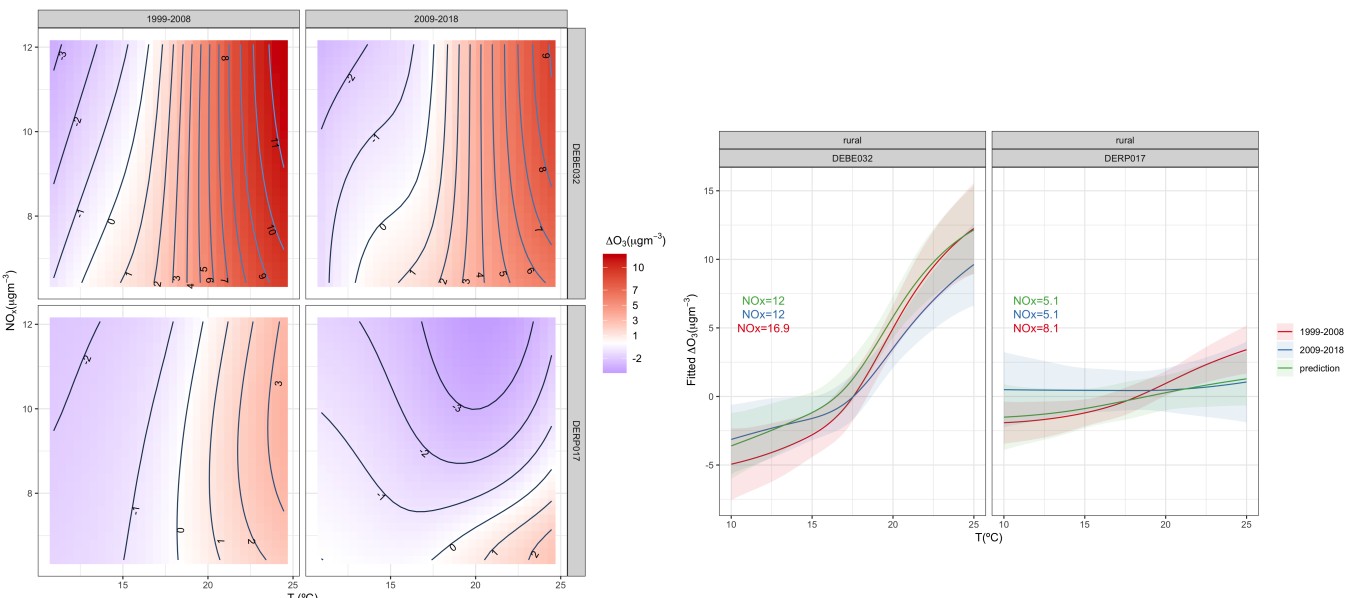

**Figure 8.** As figure 5, but for the rural stations in Berlin (DEBE032) and Rhineland-Palatinate (DERP017).

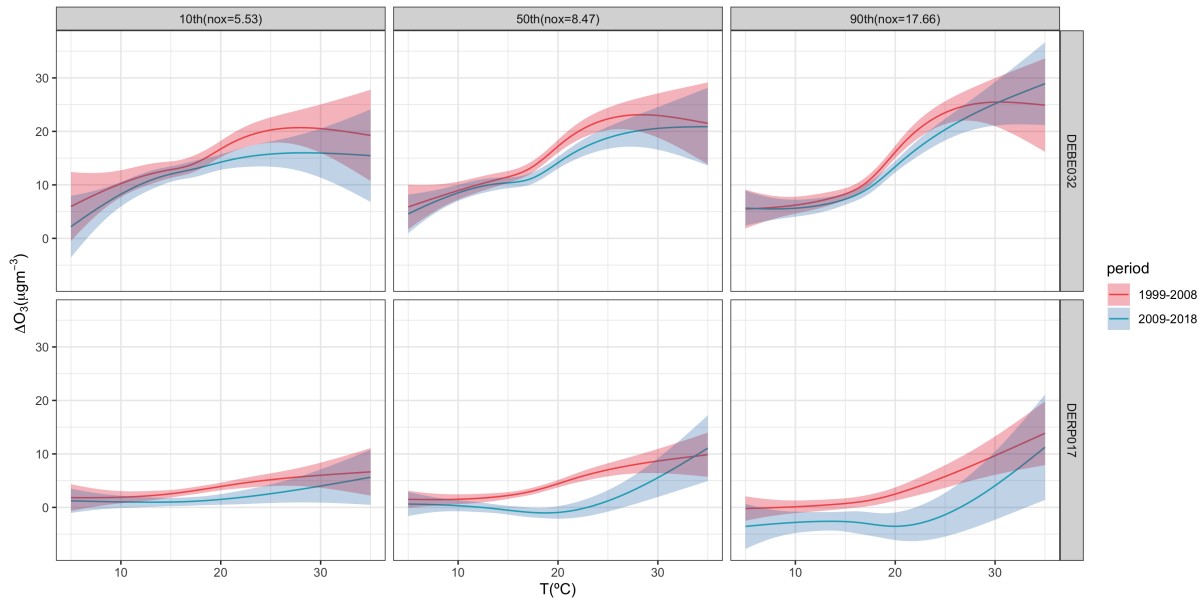

**Figure 9.** As figure 6, but for the rural stations in Berlin (DEBE032) and Rhineland-Palatinate (DERP017).





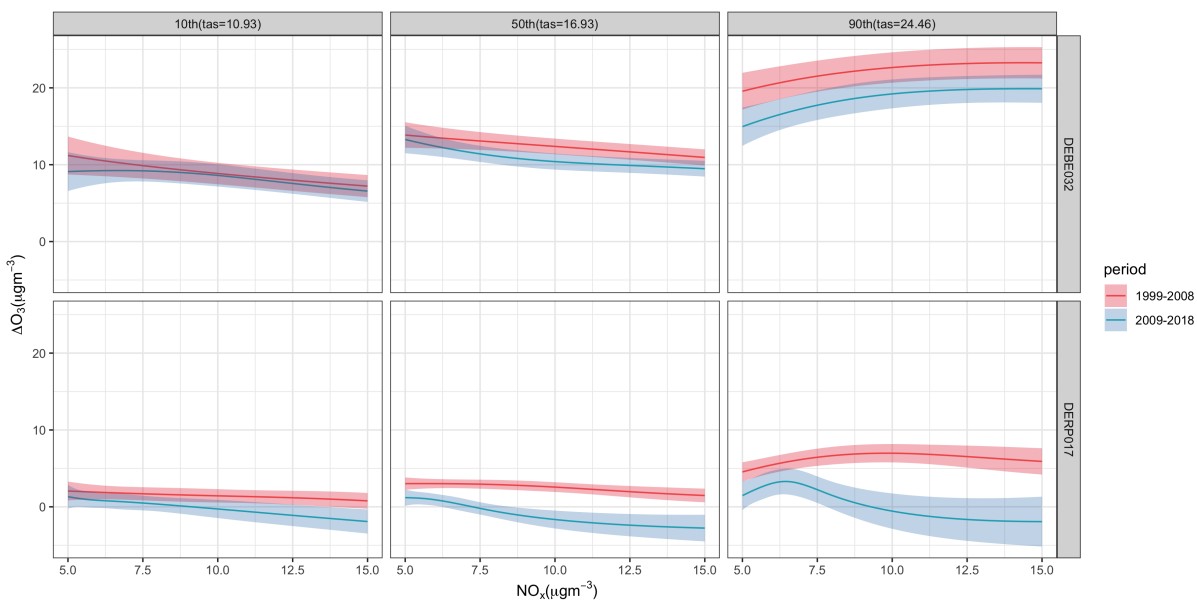

**Figure 10.** As figure 7, but for the rural stations in Berlin (DEBE032) and Rhineland-Palatinate (DERP017).

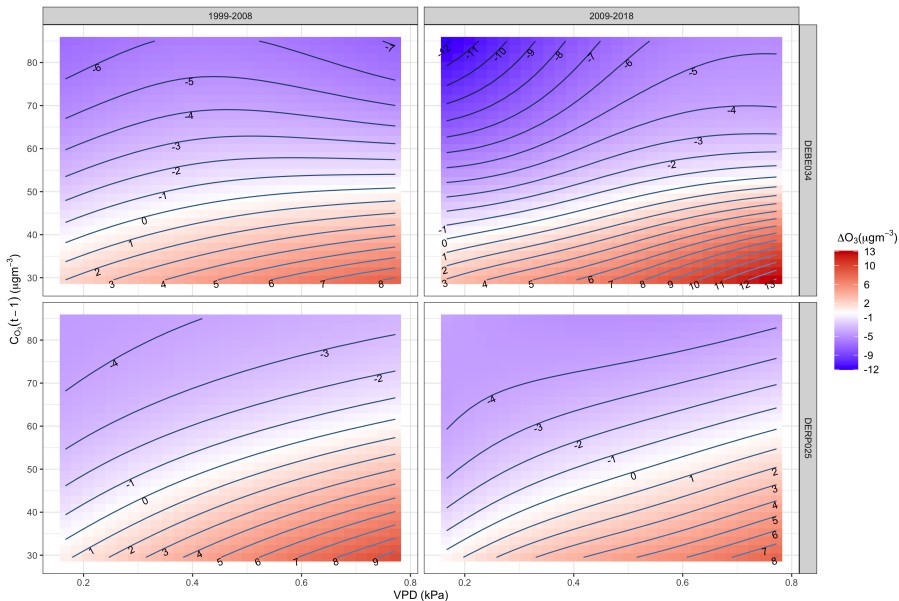

**Figure 11.** Contour plot for the interaction VPD-$C_{O_3}(t-1)$ at the urban stations in Berlin (DEBE034) and Rhineland-Palatinate (DERP025) for the first period 1999-2008 and second period 2009-2018.



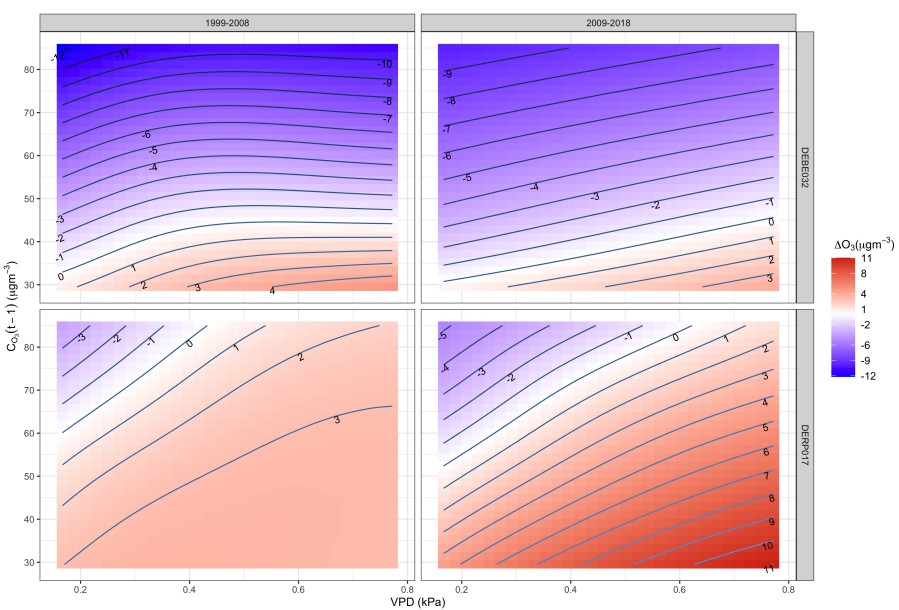

**Figure 12.** As figure 11, but for the rural stations in Berlin (DEBE032) and Rhineland-Palatinate (DERP017).

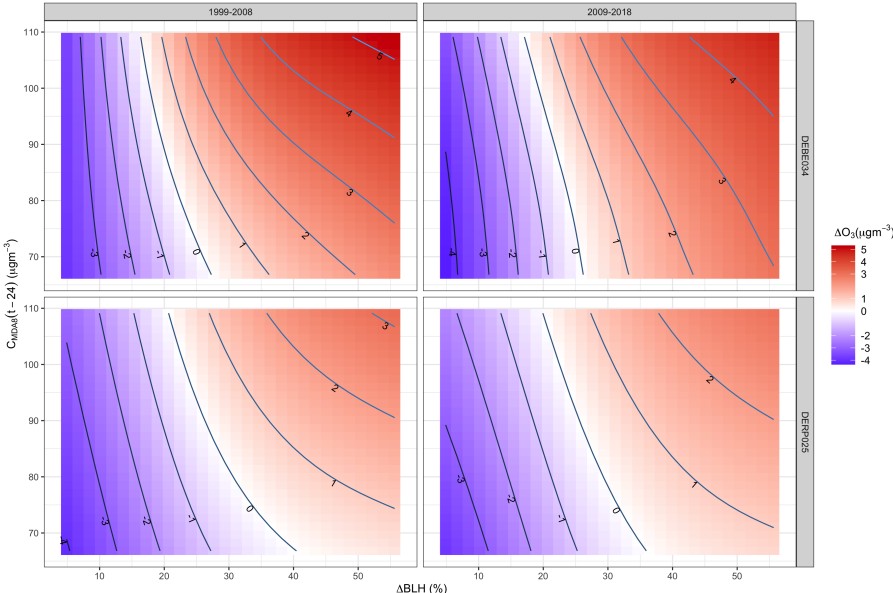

**Figure 13.** Contour plot for the interaction $\Delta$BLH-$C_{MDA8}(t-24)$ at the urban stations in Berlin (DEBE034) and Rhineland-Palatinate (DERP025) for the first period 1999-2008 and second period 2009-2018.





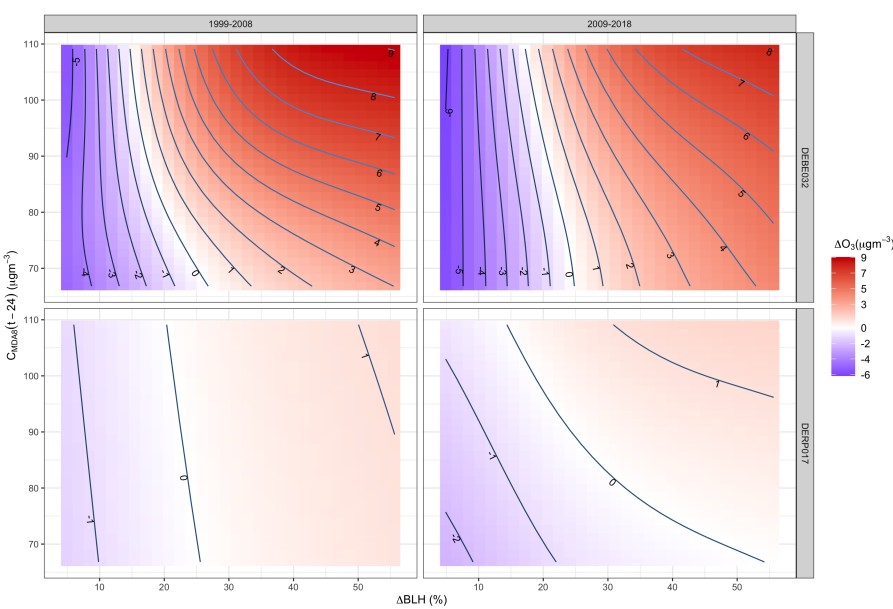

**Figure 14.** As figure 13, but for the rural stations in Berlin (DEBE032) and Rhineland-Palatinate (DERP017).



# 7    Table



**Table 1.** Median values of the covariates during the period first 1999-2008, and second period 2009-2018. Note that these values are obtained from the input data used for the GAMS (i.e. previously filtered).

| code | period | NOx | tas | BLh | VPD | lag | lag24 | type |
|---|---|---|---|---|---|---|---|---|
| DEBE032 | 1999-2008 | 14.00 | 17.17 | 23.58 | 0.35 | 30.00 | 74.00 | rural |
| DEBE032 | 2009-2018 | 12.52 | 17.36 | 24.67 | 0.37 | 27.60 | 73.15 | rural |
| DERP017 | 1999-2008 | 7.53 | 17.57 | 21.01 | 0.44 | 80.00 | 100.80 | rural |
| DERP017 | 2009-2018 | 6.39 | 16.33 | 19.49 | 0.36 | 70.90 | 96.26 | rural |
| DEBE034 | 1999-2008 | 28.00 | 17.71 | 22.65 | 0.39 | 42.50 | 76.12 | urban |
| DEBE034 | 2009-2018 | 24.26 | 18.41 | 23.36 | 0.46 | 48.84 | 79.11 | urban |
| DERP025 | 1999-2008 | 21.07 | 18.82 | 24.13 | 0.53 | 52.00 | 90.12 | urban |
| DERP025 | 2009-2018 | 15.23 | 19.00 | 24.67 | 0.56 | 51.28 | 85.46 | urban |
| DEBE051 | 1999-2008 | 16.53 | 16.76 | 22.00 | 0.32 | 40.00 | 81.62 | suburban |
| DEBE051 | 2009-2018 | 14.30 | 16.98 | 25.20 | 0.35 | 39.18 | 81.31 | suburban |
| DESN045 | 1999-2008 | 13.53 | 16.77 | 21.39 | 0.36 | 55.94 | 90.25 | suburban |
| DESN045 | 2009-2018 | 12.23 | 17.37 | 22.05 | 0.40 | 51.74 | 86.55 | suburban |

*Data availability.* Observational ozone data used in this study are available at the Airbase database of the European Environment Agency
(EEA) data service(https://www.eea.europa.eu/data-and-maps/data/aqereporting-8). The ERA5 reanalysis products are available available on
the Climate Data Store (CDS) cloud server (https://cds.climate.copernicus.eu).

*Author contributions.* TB provided the initial study idea. NO and HR designed the statistical model with the input of TB. NO prepared the
data and conducted the analysis. The manuscript was written by NO with the contributions of TB and HR.

*Competing interests.* The authors declare no competing interests.

*Acknowledgements.* This publication was financially supported by Geo.X, the Research Network for Geosciences in Berlin and Potsdam
(grant no. SO_087_GeoX). This work was hosted by IASS Potsdam, with financial support provided by the Federal Ministry of Education
and Research of Germany (BMBF) and the Ministry for Science, Research and Culture of the State of Brandenburg (MWFK).





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
