# Peer review of "Observed changes in the temperature dependence response of surface ozone under NOx reductions"

_Atmospheric Chemistry and Physics, 2020_

## Referee Comment (RC1) · Anonymous Referee #2 · 14 Sep 2020

The authors present an interesting study on the so called "climate penalty" i.e. the dependency of ozone concentration on temperature. They have analyzed a large observational dataset from different measurements station in Germany and estimated the changes of the climate penalty over time. Additionally they tried to interpret the data by using statistical method (general additive methods, GMAs). The manuscript is clear and well written.

Despite that, it is still unclear to me what the real goal of this manuscript is. If the authors are trying to explain the causes of climate penalty changes (at least for these stations), I believe the manuscript fall short of its objective. I have the impression that the authors overlooked (voluntarily or involuntarily) the importance of many factors for a clear explanation of the climate penalty.

Here below I have listed the main concerns, followed by more specific comments, as well as few technical corrections.

Main comments:

**VOC contribution:** I generally miss the consideration of VOC in this paper. Together with NOx, VOC determine the dominant chemical regime which is relevant for interpreting the changes of ΔO3 in response to decreasing NOx. For a specific time, temperature is likely a good approximation for VOC concentrations as described in the paper, but when comparing two different time periods it is important to consider the change in VOC and the resulting VOC/NOx ratio. I think some of the open questions in this paper could be solved with an evaluation of VOC data. If it is hard to find VOC data: I know most of the measurement sites in Germany measure benzene which could be analyzed as a surrogate for anthropogenic VOC for the relative development over time

At the urban sites it is pointed out that a VOC limited regime is dominant (at least for most NOx concentrations and temperatures). However, this doesn't fit together with the abstract where it is said that 'lowering NOx concentrations resulted in decreasing O3 production rates'. Decreasing NOx concentrations only lead to a decrease in O3 production in a NOx limited regime (Pusede et al. 2012).

I do additionally wonder if any changes in biogenic emissions could be expected. Although temperature (as mentioned before) can be used for VOC concentrations, I wonder if we can consider this dependency stable over the entire observed period (or even within the two periods in which the data are divided). Maybe the absolute changes are not significant, but almost certainly is their relative impact in comparison with anthropogenic VOC emissions.

**Figure interpretation:** Although it does make sense that the temperature sensivity of O3 production decreases for decreasing NOx, I don't think the Figures show this. For the urban site in RP (Rhineland Palatinate) I get the impression that the slope in Figure 5 right (and therefore the climate penalty) is higher for the second time period, which would indicate the opposite.

**Results representativeness:** The title of the paper might be a bit too general considering that the analysis focuses on a small number of locations in Germany. Again, the response of O3 to NOx reductions can have different outcomes depending on VOC and the dominant chemical regime which varies strongly with the considered location

**GAMs model:** It would be great if a little more background information could be provided on how the calculations were made and how the model performance was tested. As mentioned later, the model description could be moved entirely to the electronic supplement and also enlarged to include a more exhaustive description, while on the manuscript only a small summary could be shown. Importantly, the code availability and the data source should be mentioned (best if the code could be uploaded as supplement as well).

Specific comments

**Page 1, line 11** : nice that with the GAMs analysis you finds exactly the same results as before (see line 6) (i.e. „decreasing sensitivity of temperature" in the second period). Instead of repeating the same sentence as a new finding, you could add "the GAMs model confirm that…"

**Page 1, Line 14 :** Please consider the main comments: of course NOx concentration is not the only factor influencing the climate penalty. This sentence is indeed too general for explaining your 23 pages work!

**Page 1, line 20 : (**Sillman, 1999) :  The authors of this paper shown in the references are Pusede et al. (2015)

**Page 1, lines 23-24 :** What about anthropogenic VOC? Many are temperature-dependent, too. Compare Pusede et al. 2014 (also relevant for the summary)

**Page 2, line 33-34 :** Jing et al. (2017) state a value of 0.43 ppb $K^{-1}$ by which the climate penalty was lower in 1999-2007.

**Page 3, line 24**  : '… variability of O3 production can explain a considerable proportion…' Do you have any reference?

**Page 4, equation 1 :** Please explain all variables and what the function means. To me it seems a linear regression (as you mentioned a linear model) applied for the 2 periods. Do we really need to define a mask (P) and 2 equations to explain that? I have the impression that here the readability has been somehow compromised and made difficult to follow.

**Page 6, line 9-31 :** The description of the GAMs model is indeed difficult to follow. Most possibly is because the needed background in formations are quite large. Although the normal reader would just trust the results, I think that the results of this manuscript should be reproducible by anyone interested (I think that ACP has also a policy on that, if I'm not wrong). Therefore I would strongly suggest to move the GAMs model description to the supplement, with the goal to expand it and make it intelligible by anyone interested. Further, in the electronic supplement of the model the code used for the analysis (i.e. the GAMs algorithm) should be make publicly available, as well with an indication where to download (or obtain) the station's data.

**Page 7, line 4 :** Why did you need to fit the GAMs model if you build exactly from the dataset? I thought this information to be already used in the construction of the model.

**Page 7, line 19  Figure S1:**  I find this plot hard to analyze. Maybe it could help to add a black stroke color to all data points.

**Page 7, line 26  Figure 3:** For Figure S3, I like that the differences between the two periods become clear on first sight, maybe this could be added to Figure 3 for a better understanding.

**Page 8, line 11  Figure 4:** Could you make a comparison plot similar to Figure S3 for Figure 4, too?

**Page 8, line 4 :** "A final model including three interaction terms" : Again, the description of the GAMs model was not exceptional. Probably would not hurt if you list again these terms here.

**Page 9, line 8:**             '…implying a VOC-limited chemistry'

If I understand the plot correctly, for DERP025 for low NOx and high temperatures the chemistry is NOx-limited for example at T=24°C for the first time period 1999-2008, ΔO3 increases from 5 to

6 µg/m³ and then decreases again, so the change from VOC to NOx limited chemistry would occur at c(NOx)=20 µg/m³. For the second time period 2009-2018 and the same temperature T=24°C the regime change occurs already at c(NOx)=15 µg/m³. For DEBE034 it looks like an even smaller part of each plot shows a NOx limited regime. I think this could be a result from lower NOx at DEPR025 compared to DEPR034 and a more effective NOx reduction over time (as shown on the right). Maybe this could be interesting to mention or to further analyze.

**Page 9, line 9** : 'We found a stronger temperature dependence of ΔO3 in the first period…'

In Figure 5 left, I have problems to identify this, e.g. for DERP025 for c(NOx)=20 µg/m³, for a temperature increase from T1=20°C to T2=24°C, ΔO3 increases from 2 to 6 µg/m³ for 1999-2008 and from 0 to 6 µg/m³ for 2009-2018, same for higher NOx. Figure 5 right, too, doesn't support the stronger temperature dependence in the first period: For Rhineland Palatinate I would even say that the correlation of temperature and ΔO3 is larger for the second period when looking at the ΔO3 for a set temperature interval, for example 20-25°C: For 1999-2008, ΔO3 increases by approx. 6.5 µg/m³ and for 2009-2018, ΔO3 increases by approx. 7.5 µg/m³.

For Berlin, the temperature dependence could be slightly stronger for the first period, but the difference is marginal and hard to tell from the graph.

**Page 9, lines 21-27 :** What about the influence of VOC? If you look at a certain time the influence of VOC can be well represented by temperature, but between the two different time periods emission controls have decreased VOC. For the shown temperature interval and the mean NOx concentration the dominant chemical regime is VOC limited. If NOx reductions exceed VOC reductions O3 increases which is shown by the prediction in Figure 5 left. However, if VOC reductions are larger than expected (and if I get this right the prediction only considers a change in NOx, not in VOC), this would counter run the O3 increase made by the prediction (because decreasing VOC decrease O3 in a VOC limited regime, Pusede et al. 2012) and would yield a lower curve than the predicted one. So maybe VOC reductions were more effective in Rhineland Palatinate than in Berlin – just an idea.

**Page 10, line 2-3 :** 'This indicates that the ΔO3 sensivity to higher NOx at moderate and high T is lower in the second period.'

…or indicates that VOC reductions at RP were more effective. (which would be consistent with the deviating prediction line for Rhineland Palatinate in Figure 5 right)

**Page 10, line 5** : '… temperature dependence of ΔO3 is stronger in the first period…'

I do see this here for the rural sites, but not for the presented urban sites. Higher NOx in urban areas could be an explanation here.

**Page 10, line 12 :** '… could partially explain the decrease in ΔO3 estimates …'

In Berlin, for high temperatures a NOx limited regime is dominant for any NOx concentration. For lower temperatures a VOC limited regime is dominant. I think this is a good explanation for the observed plot course in Figure 8 left. The red and the blue plot intersect at around 17°C which is approximately the transition temperature for the observation of a NOx (above) and a VOC limited regime (below). For lower NOx during the second period ΔO3 should therefore be lower in the NOx limited regime at high temperatures but higher in the VOC limited regime at low temperatures.

**Page 10, line 25 :** Figure 10. For high temperatures, both periods in Berlin and the earlier period in RP are dominated by a NOx limited chemistry which is also shown by Figure 8 left. Consequently, ΔO3 increases with increasing NOx. For the second period in RP a VOC limited regime is dominant and ΔO3 decreases for increasing NOx. I find this surprising considering the shown decrease in mean NOx in

Figure 8 right from the first to the second period. Could there be an explanation? What about VOC concentrations at these sites?

**Page 11, Section 4.3.2 :** Which parameters could have changed over time so that the plots look different for the first and the second period particularly for RP and why not for Berlin? It might be better for the understanding of the results if the explanation of VPD was included a bit earlier, maybe at the beginning of this paragraph.

Technical corrections

| Page 1, line 22 | varies |
| Page 2, line 13,22,24 | the United States |
| Page 2, line 21 | photochemical |
| Page 3, line 7 | maximum |
| Page 5, line 5 | the latter |
| Page 9, line 10 | the second period |
| Page 9, line 21 | shown |
| Page 10, line 1/36,26 | decrease |
| Page 10, line 2 | indicates |
| Page 12, line 2 | decrease |
| Page 12, line 11 | lead |

---

## Referee Comment (RC2) · Anonymous Referee #1 · 19 Sep 2020

In this work the authors examine longterm O3 data from surface stations in Germany, comparing hourly ozone changes with various ambient conditions such as temperature and NOx levels. Using generalized additive models across two halves of the total temporal domain (1999-2018) to model and combine the influence of these driving factors, the authors conclude that these two time periods show differences in the O3-temperature relationship, driven only in part by observed NOx emissions reductions over that period.

While the topic of pollution production and its atmospheric influences is important and complex, I think this manuscript needs considerable development to be considered a novel and meaningful contribution to the existing literature on the subject. In particular, I have the following concerns:

[Figure]

- The choice of temporal division (analyzing the full time series in two evenly divided chunks) strikes me as arbitrary and problematic. Unless the year 2009 has some special significance that is not discussed in the text, I see no reason to set up the binary comparisons between time periods as performed here. The division is ostensibly made to compare a higher NOx time period (1999-2008) to a lower NOx period (2009-2018), but not only is this assumption not necessarily valid for all stations during all years (see Figure 2), it also neglects the wide variety of other changes that may have occurred over the two decade span that could influence ozone and its relationships with ambient conditions. Compared to other methods of distinguishing between higher and lower NOx conditions (for example, by leveraging the so-called weekend effect), comparing consecutive decades individually and ascribing their differences to only one factor (NOx emissions) strikes me as flawed. The authors' observation that "decreasing NOx concentrations are not the only factor causing the observed changes" underscores this fact, and raises the question of why they chose to dissect their long term data set in this fashion at all. I would recommend rethinking the approach here, and identifying a methodology that is less subject to non-stationarities in external variables.

- On a related note, while this study considers an assortment of ozone-influencing covariates alongside temperature and NOx, it conspicuosly ignores others. For example, VPD is considered to represent dry deposition rates, and temperature is identified as a surrogate for biogenic emissions, but there is no mention of changes in the plants responsible for these effects in the first place. Changes in land cover, whether in the form of ongoing biosphere growth and aging, losses due to anthropogenic land development, or shifts in plant speciation can all have drastic impacts on biogenic emissions, their temperature dependence, and other surface/atmosphere connections such as ozone deposition velocities. It is surprising, therefore, to see no model inclusion or even mention of how changes across the temporal domain could influence O3-temperature dependence in this study.

- The primary conclusions of this paper are generally either unsurprising and underdeveloped. The correlation between NOx emissions and the O3 climate penalty has been consistently observed, modeled, and dissected in studies performed all over the world, and there doesn't seem to be much added to the conversation here. Furthermore, areas of potential interest, such as the observation that "NOx reductions alone can not explain the changes in the temperature dependence of O3" go largely unexplained, leaving open the questions that could lead to more significant and meaningful answers.

- Figure and text quality are highly inconsistent, with some glaring issues scattered throughout. Puzzling color and layout choices make it difficult to make sense of visualizations. For example, Figure 1 includes a color scheme to show station altitude, but these colors show no obvious consecutive progression, making the ready comparison of sites awkward and unintuitive. Panels of contour and ribbon plots might show features of interest, but, aside from textual description of very basic features, they don't receive much development or interpretation in the text. Grammar, spelling, and phrasing mistakes often impede manuscript fluency and flow.

- Data filtering seems to be extremely strict, and it is unclear how this filtering process may itself have resulted in spatiotemporal differences. Were there any discernable patterns with respect to the percentage of hours kept for analysis across station and year? Could changes in the frequency of removed hours over time, or between stations, confound comparisons? This seems like a potentially large source of statistical artifacts, if not examined and accounted for.

- Model selection deserves more attention and description. It is stated that the goal was "a common model well defined across all of the stations", but later it is mentioned that "the model selection procedure was applied separately at each station and period." Does this mean that forward selection was performed individually by station and time? If so, this is a major problem in the interpretation of model output. If not, it's unclear how these two statements are reconconciled. How was forward selection applied in a way that resulted in a common model across all stations, while also being applied

separately by station and period?

All in all, I think that this manuscript contains the foundation for an interesting paper, but that it is not there yet. I recommend that it be rethought and far more significantly developed, in particular addressing some of these concerns, before being considered for publication again.
* * *

---

## Author Comment (AC1) · 3 Nov 2020

**Response to referee comments on acp-2020-691**

Noelia Otero, Henning W. Rust and Tim Butler

**General comment**

We would like to thank both referees for their constructive and useful comments, which helped to significantly improve the manuscript. We have carefully revised the manuscript according to the major concerns and the specific comments. Here, we provide our responses. In each case, we have copied the referees comments (in bold) and our responses are written in standard script. While we have tried to balance preservation of as much of the original text as possible, we have substantially modified some parts of the text to clarify and improve the presentation of the results following the suggestions of both referees. We consider that the changes made in response to the referees comments helped to improve considerably the manuscript and we hope the editor and referees find the revised version suitable for publication in ACP. We also append a marked-up version of the manuscript with the changes mentioned in our responses to the referees. Text deleted is shown as cross out sentences, and extra or new text in red script.

**Response to Referee #1**

The authors present an interesting study on the so called "climate penalty" i.e. the dependency of ozone concentration on temperature. They have analyzed a large observational dataset from different measurements station in Germany and estimated the changes of the climate penalty over time. Additionally they tried to interpret the data by using statistical method (general additive methods, GAMs). The manuscript is clear and well written. Despite that, it is still unclear to me what the real goal of this manuscript is. If the authors are trying to explain the causes of climate penalty changes (at least for these stations), I believe the manuscript fall short of its objective. I have the impression that the authors overlooked (voluntarily or involuntarily) the importance of many factors for a clear explanation of the climate penalty.

The primary objective of this study is to analyze how  $NO_x$  reductions have influenced the temperature dependence of  $O_3$ . As stated in the introduction, the temperature dependence of  $O_3$  is complex and varies in space and time due to differing chemical and meteorological mechanisms that influence  $O_3$  formation (Pusede et al. 2015). Previous studies have reported a weaker  $O_3$  sensitivity to temperature over the past years, likely due to reduction of emissions of  $O_3$  precursors. Here we specifically focus on the impacts of  $NO_x$  reductions that have drastically declined for last past decades in Germany. Ultimately, we aimed to infer changes in VOC accompanying  $NO_x$  reductions that might contribute to the changes in the temperature dependence of  $O_3$ . We have stressed the main objectives of the study in the revised version of the manuscript.

Here below I have listed the main concerns, followed by more specific comments, as well as few technical corrections.

**Main comments:**

VOC contribution: I generally miss the consideration of VOC in this paper. Together with NOx, VOC determine the dominant chemical regime which is relevant for interpreting the changes of  $\Delta O_3$  in response to decreasing NOx. For a specific time, temperature is likely a good approximation for VOC concentrations as described in the paper, but when comparing two different time periods it is important to consider the change in VOC and the resulting VOC/NOx ratio. I think some of the open questions in this paper could be solved with an evaluation of VOC data. If it is hard to find VOC data: I know most of the measurement sites

**in Germany measure benzene which could be analyzed as a surrogate for anthropogenic VOC for the relative development over time.**

We agree with Referee #1 that VOC changes are relevant as well as local chemical regimes, which are defined based on the VOC/NOx ratios. It was not our intention to dismiss the contribution of VOC, on the contrary, as mentioned in our earlier comment, we aim to provide further insights into changes in VOC over the past decades through the observed changes in the temperature sensitivity of  $O_3$  under  $NO_x$  reductions.

Following the referee suggestion, we have specifically looked at long-term benzene data over Germany. We have extracted all available benzene data from 1999-2018 from Airbase ((https://www.eea.europa.eu/data-and-maps/data/aqereporting-8). Unfortunately, only two stations from the 29 sites analysed in our study presented data (Fig.R1). In Fig.R1 we show the annual averages of benzene in summertime (here defined as July-August-September) for the stations of the study. From Fig.R1 we see a downward trend for in both stations. However, it was not possible to objectively examine trends for the rest of the stations of the study.

Figure R1.Annual average of benzene measurements for two urban stations included in the presented manuscript.

We have further assessed the measurements of benzene for the rest of available stations (different from the stations analysed in this study). Figure R2 shows the annual averages for the available stations and years. As in Fig.R1, it can be observed a general downward trend, but the quality of the data, in terms of both temporal and spatial coverage, was not sufficient for this study.

Figure R2. Annual average of benzene measurements for the rest of the stations .

Therefore, as stated in the manuscript and given that long-term records of VOC are generally not available, we used temperature that has been shown to be an useful proxy for representing VOC (LaFranchi et al. 2011, Pusede et al. 2014).

As pointed out by Referee #1 changes in the ozone chemistry due to changes in the ratio of VOC/NOx emissions is a key point when comparing two different time periods. In response to another comment from Referee #2, we have examined the weekend-weekday effect, which can be used as an indicative of the dominant chemistry regime (Steiner et al. 2006). Furthermore, we have now added the weekend-weekday effect in the revised version of the manuscript. For more details, please see our response to first comment from Referee #2 and Figs.R3 and R4.

At the urban sites it is pointed out that a VOC limited regime is dominant (at least for most NOx concentrations and temperatures). However, this doesn't fit together with the abstract where it is said that 'lowering NOx concentrations resulted in decreasing O3 production rates'. Decreasing NOx concentrations only lead to a decrease in O3 production in a NOx limited regime (Pusede et al. 2012).

Referee #1 is correct. The urban stations of this study show a dominant VOC-limited chemistry (see Fig. 4), under which decreasing VOC would be more effective in reducing  $O_3$ . We have modified accordingly the abstract.

I do additionally wonder if any changes in biogenic emissions could be expected. Although temperature (as mentioned before) can be used for VOC concentrations, I wonder if we can consider this dependency stable over the entire observed period (or even within the two periods in which the data are divided). Maybe the absolute changes are not significant, but almost certainly is their relative impact in comparison with anthropogenic VOC emissions.

We agree, and we also hypothesize that changes in biogenic emissions have contributed to the temperature dependence of  $O_3$ . In particular, soil moisture deficit is a relevant factor of stress for isoprene emissions and when soil moisture is limited plants decrease their emissions of isoprene (Guenther et al. 2006). Severe droughts might influence plant growth and limit biomass production, which can lead to a reduction of isoprene emission (Emmerson et al. 2019). Moreover, earlier studies suggested that plants have very different drought responses, and while isoprene emission tend to decrease with low levels of soil moisture below a certain value, this decreasing might be less significant under exteded severe drought (Pegoraro et al. 2004). In addition to that, recent studies have shown the regional hot and dry conditions over last decade in central Europe

(Buras et al. 2019, Ionita et al. 2017), which might have a significant impact on biogenic emissions (e.g. severe droughts reduce stomatal uptake of ozone and its precursors (Demetillo et al. 2019)). Thus, it is reasonable to expect changes in biogenic emissions. We have added some extra text in the introduction discussing the feedbacks from vegetation, including the effect of soil moisture.

Figure interpretation: Although it does make sense that the temperature sensivity of O3 production decreases for decreasing NOx, I don't think the Figures show this. For the urban site in RP (Rhineland Palatinate) I get the impression that the slope in Figure 5 right (and therefore the climate penalty) is higher for the second time period, which would indicate the opposite.

Yes, you are right, thank you. From the Figure 5 (showed in the old version of the manuscript) we see a general decrease of  $\Delta O_3$  during the second period 2009-2018 (lower NOx) when comparing to the estimates in the first period 1999-2008. This feature is well observed in Berlin. However, in Rhineland Palatinate it can be observed that  $\Delta O_3$  tends to decrease for NOx > 20  $\mu$ gm-3 at all temperature ranges, but increases at higher temperatures (>20 °C) when lowering NOx (<20  $\mu$ gm-3). This is likely due to changes in VOC that would also explain the differences observed in the shapes of the regression lines when comparing the prediction (green line) and GAM-P2 (blue line). We have taken note of this comment and we have stressed the influence of VOC in the case of Rhineland Palatinate.

**Results representativeness: The title of the paper might be a bit too general considering that the analysis focuses on a small number of locations in Germany. Again, the response of O3 to NOx reductions can have different outcomes depending on VOC and the dominant chemical regime which varies strongly with the considered location**

We would like to highlight that the selection of the area of this study was mainly motivated by the availability of long-term records of co-located data. However, we understand this comment, since the outcomes of this study are not easily extrapolated. Thus, we have decided to modify slightly the title as:

"Observed changes in the temperature dependence response of surface ozone under NOx reductions over Germany"

GAMs model: It would be great if a little more background information could be provided on how the calculations were made and how the model performance was tested. As mentioned later, the model description could be moved entirely to the electronic supplement and also enlarged to include a more exhaustive description, while on the manuscript only a small summary could be shown. Importantly, the code availability and the data source should be mentioned (best if the code could be uploaded as supplement as well).

We have used the standard tools (e.g. QQ-plot, histograms) to evaluate the model fit and we assessed the model performance by using the  $\mathbb{R}^2$ . Following your recommendation we provide now a more detailed model description in the supplement material. We have added the code availability in the revised version of the manuscript.

**Specific comments**

Page 1, line 11 : nice that with the GAMs analysis you finds exactly the same results as before (see line 6) (i.e. "decreasing sensitivity of temperature" in the second period). Instead of repeating the same sentence as a new finding, you could add "the GAMs model confirm that..."

The sentence has been changed.

Page 1, Line 14 : Please consider the main comments: of course NOx concentration is not the only factor influencing the climate penalty. This sentence is indeed too general for explaining your 23 pages work!

We have modified the manuscript accordingly with responses and extra analysis provided here.

Page 1, line 20 : (Sillman, 1999) : The authors of this paper shown in the references are Pusede et al. (2015)

The reference has been corrected.

Page 1, lines 23-24 : What about anthropogenic VOC? Many are temperature-dependent, too. Compare Pusede et al. 2014 (also relevant for the summary)

We have added some extra text to mention the role of VOC temperature-dependent.

Page 2, line 33-34 : Jing et al. (2017) state a value of 0.43 ppb K-1 by which the climate penalty was lower in 1999-2007.

We have corrected this thank you.

Page 3, line 24 : '... variability of O3 production can explain a considerable proportion...' Do you have any reference?

We have added the corresponding reference here (Pusede et al. 2015).

Page 4, equation 1 : Please explain all variables and what the function means. To me it seems a linear regression (as you mentioned a linear model) applied for the 2 periods. Do we really need to define a mask (P) and 2 equations to explain that? I have the impression that here the readability has been somehow compromised and made difficult to follow.

Yes, we use a linear regression model. We introduced a categorical variable (period,P) to assess the significant differences between the slopes. As an alternative, a significant t-test can be applied to examine the significant differences between the coefficients (slopes). To avoid complexity, we have now simplified equation 1) and assessed differences using a significant t-test. As stated now in the revised version of the manuscript Equation 1) has the following form:

$$Y(t) = a + m_{O_3T}T(t) + \epsilon(t) \tag{1}$$

with  $\epsilon(t) \sim \mathcal{N}(0, \sigma^2)$ ,  $\alpha$  being the constant offset and  $m_{O_3T}$  the slope of the linear relation.Y(t), T(t) are the time series of MDA8 and daily maximum temperature (respectively).

Page 6, line 9-31 : The description of the GAMs model is indeed difficult to follow. Most possibly is because the needed background in formations are quite large. Although the normal reader would just trust the results, I think that the results of this manuscript should be reproducible by anyone interested (I think that ACP has also a policy on that, if I'm not wrong). Therefore I would strongly suggest to move the GAMs model description to the supplement, with the goal to expand it and make it intelligible by anyone interested. Further, in the electronic supplement of the model the code used for the analysis (i.e. the GAMs algorithm) should be make publicly available, as well with an indication where to download (or obtain) the station's data.

We agree that the description of the GAM is not straightforward. After carefully revised section 3.3 we have modified some parts of the text in order to better explain the basics of GAM. Moreover, we have moved some text to the Supplement Material and following the suggestion from this referee we have also extended the model description. As stated in the data availability, the air quality data can be extracted from Airbase (https://www.eea.europa.eu/data-and-maps/data/aqereporting-8) and the meteorological variables from Climate Data Store (CDS) cloud server (https://cds.climate.copernicus.eu). The code availability has been added along with the data availability in the revised version.

**Page 7, line 4 : Why did you need to fit the GAMs model if you build exactly from the dataset? I thought this information to be already used in the construction of the model.**

As mentioned in the earlier comment, we have moved part of the text to the Supplement. We would like to clarify to the referee that based on the selection procedure, we selected a final GAM and then, we build a GAM individually for each station and period.

Page 7, line 19 Figure S1: I find this plot hard to analyze. Maybe it could help to add a black stroke color to all data points.

The plot has been updated in the revised version of the supplement.

Page 7, line 26 Figure 3: For Figure S3, I like that the differences between the two periods become clear on first sight, maybe this could be added to Figure 3 for a better understanding.

We have added Fig. S3 to the main text.

Page 8, line 11 Figure 4: Could you make a comparison plot similar to Figure S3 for Figure 4, too?

Figure 4 shows the deviance explained  $(R^2)$  obtained from GAM-P1 and GAM-P2 built from different datasets. Thus, it is not possible to establish a similar comparison as done for Figure S3 (old version).

Page 8, line 4 : "A final model including three interaction terms" : Again, the description of the GAMs model was not exceptional. Probably would not hurt if you list again these terms here.

We have added extra text in the revised version of the manuscript.

Page 9, line 8: '...implying a VOC-limited chemistry' If I understand the plot correctly, for DERP025 for low NOx and high temperatures the chemistry is NOx-limited for example at T=24°C for the first time period 1999-2008,  $\Delta O_3$  increases from 5 to 6 ug/m3 and then decreases again, so the change from VOC to NOx limited chemistry would occur at c(NOx)=20 ug/m3. For the second time period 2009-2018 and the same temperature T=24°C the regime change occurs already at c(NOx)=15 ug/m3. For DEBE034 it looks like an even smaller part of each plot shows a NOx limited regime. I think this could be a result from lower NOx at DEPR025 compared to DEBE034 and a more effective NOx reduction over time (as shown on the right). Maybe this could be interesting to mention or to further analyze.

Thank you for this useful comment. As Referee #1 points out, at Rhineland-Palatinate the change in the chemistry to  $NO_x$ -limited at higher temperatures occurs at lower values of  $NO_x$  for the second period. From Fig. R4 (please see in the responses to Referee #2) it can observed that the weekend-weekday effect at Rhineland-Palatinate during the second period has significantly decreased, which suggests a transition to a  $NO_x$ -limited system, while in Berlin the weekend-weekday effect is similar in both periods. Consistently, we found in Berlin a general VOC-limited regime in first and second period, which indicate that further reductions should be required for mitigating the impacts of warmer temperatures (i.e. climate penalty). This results also point out VOC reductions over time. We have emphasized it in the revised version of the manuscript.

Page 9, line 9 : 'We found a stronger temperature dependence of  $\Delta O_3$  in the first period...' In Figure 5 left, I have problems to identify this, e.g. for DERP025 for  $c(NOx)=20 \text{ ug/m}^3$ , for a temperature increase from T1=20°C to T2=24°C,  $\Delta O_3$  increases from 2 to 6 ug/m3 for 1999-2008 and from 0 to 6 ug/m3 for 2009-2018, same for higher NOx. Figure 5 right, too, doesn't support the stronger temperature dependence in the first period: For Rhineland Palatinate I would even say that the correlation of temperature and  $\Delta O_3$  is larger for the second period when looking at the  $\Delta O_3$  for a set temperature interval, for example 20-25°C: For 1999-2008,  $\Delta O_3$  increases by approx. 6.5 ug/m3 and for 2009-2018,  $\Delta O_3$  increases by approx. 7.5 ug/m3.For Berlin, the temperature dependence could be slightly stronger for the first period, but the difference is marginal and hard to tell from the graph.

We have carefully revised the text and taken note of this comment. We agree with the referee that the dependence with temperature is not significantly larger in the first period when comparing to the second period. We have modified the text accordingly.

Page 9, lines 21-27 : What about the influence of VOC? If you look at a certain time the influence of VOC can be well represented by temperature, but between the two different time periods emission controls have decreased VOC. For the shown temperature interval and the mean NOx concentration the dominant chemical regime is VOC limited. If NOx reductions

exceed VOC reductions O3 increases which is shown by the prediction in Figure 5 left. However, if VOC reductions are larger than expected (and if I get this right the prediction only considers a change in NOx, not in VOC), this would counter run the O3 increase made by the prediction (because decreasing VOC decrease O3 in a VOC limited regime, Pusede et al. 2012) and would yield a lower curve than the predicted one. So maybe VOC reductions were more effective in Rhineland Palatinate than in Berlin – just an idea.

Yes, we agree with the referee. As mentioned in an earlier comment to this referee related to the **Figure interpretation**, from Fig. 5 (rigth) it can be noted the differences between the shapes when comparing the regression lines from the prediction and GAM-P2 in Rhineland Palatinate, which could be explained by changes in the VOC. We have stressed this in the revised verion of the manuscript, thank you.

Page 10, line 2-3 : 'This indicates that the  $\Delta O_3$  sensivity to higher NOx at moderate and high T is lower in the second period.' ... or indicates that VOC reductions at RP were more effective. (which would be consistent with the deviating prediction line for Rhineland Palatinate in Figure 5 right)

That is certainly plausible and consistent with the results showed at Fig.5. We have added some text to emphasize this in the revised version of the manuscript.

Page 10, line 5 : '... temperature dependence of  $\Delta O_3$  is stronger in the first period...' I do see this here for the rural sites, but not for the presented urban sites. Higher NOx in urban areas could be an explanation here.

As mentioned in a earlier related comment to this referee we agree that the dependence with temperature is not significantly larger in the first period at the urban stations, then this sentence has been also modified.

Page 10, line 12 : '... could partially explain the decrease in  $\Delta O_3$  estimates ... ' In Berlin, for high temperatures a NOx limited regime is dominant for any NOx concentration. For lower temperatures a VOC limited regime is dominant. I think this is a good explanation for the observed plot course in Figure 8 left. The red and the blue plot intersect at around 17°C which is approximately the transition temperature for the observation of a NOx (above) and a VOC limited regime (below). For lower NOx during the second period  $\Delta O_3$  should therefore be lower in the NOx limited regime at high temperatures but higher in the VOC limited regime at low temperatures.

We have taken note of this comment and we have added some extra text in the revised version, thank you.

Page 10, line 25 : Figure 10. For high temperatures, both periods in Berlin and the earlier period in RP are dominated by a NOx limited chemistry which is also shown by Figure 8 left. Consequently,  $\Delta O_3$  increases with increasing NOx. For the second period in RP a VOC limited regime is dominant and  $\Delta O_3$  decreases for increasing NOx. I find this surprising considering the shown decrease in mean NOx in Figure 8 right from the first to the second period. Could there be an explanation? What about VOC concentrations at these sites?

Thank you for this comment. We would like to clarify that the contours shown in Fig. 8 are limited to a range of data sufficiently supported by the observations (Section 4.3). In the case of Rhineland Palatinate we did not show the contours for low  $NO_x$  concentrations. Moreover, it is important to note that by using the  $NO_x$  filter (> 5  $\mu$ gm-3) we have limited the analysis to the space of higher  $NO_x$  concentrations. As stated in the manuscript, this filter was applied due to observed lack of low values of  $NO_x$  for some stations. To be consistent in our analysis among stations and periods we decided to apply the same filter to all stations.

From Fig. 8, at Rhineland Palatinate we observed for the second period that the  $\Delta O_3$  peak occurs at lower  $NO_x(>8\mu gm^{-3})$  than the peak observed in the first period (<8  $\mu gm^{-3}$ ), which points out the effective  $NO_x$  reductions.

Page 11, Section 4.3.2 : Which parameters could have changed over time so that the plots look different for the first and the second period particularly for RP and why not for Berlin?

It might be better for the understanding of the results if the explanation of VPD was included a bit earlier, maybe at the beginning of this paragraph.

We have now introduced the VPD and its relevance for ozone production at the beginning of the section.

Technical corrections

Page 1, line 22 varies Page 2, line 13,22,24 the United States Page 2, line 21 photochemical Page 3, line 7 maximum Page 5, line 5 the latter Page 9, line 10 the second period Page 9, line 21 shown Page 10, line 1/36,26 decrease Page 10, line 2 indicates Page 12, line 2 decrease Page 12, line 11 lead

All technical corrections have been fixed.

**Response to Referee #2**

In this work the authors examine long-term  $O_3$  data from surface stations in Germany, comparing hourly ozone changes with various ambient conditions such as temperature and  $NO_x$ levels. Using generalized additive models across two halves of the total temporal domain (1999-2018) to model and combine the influence of these driving factors, the authors conclude that these two time periods show differences in the O3-temperature relationship, driven only in part by observed  $NO_x$  emissions reductions over that period. While the topic of pollution production and its atmospheric influences is important and complex, I think this manuscript needs considerable development to be considered a novel and meaningful contribution to the existing literature on the subject. In particular, I have the following concerns:

The choice of temporal division (analyzing the full time series in two evenly divided chunks) strikes me as arbitrary and problematic. Unless the year 2009 has some special significance that is not discussed in the text, I see no reason to set up the binary comparisons between time periods as performed here. The division is ostensibly made to compare a higher  $NO_x$  time period (1999-2008) to a lower  $NO_x$  period (2009-2018), but not only is this assumption not necessarily valid for all stations during all years (see Figure 2), it also neglects the wide variety of other changes that may have occurred over the two decade span that could influence ozone and its relationships with ambient conditions. Compared to other methods of distinguishing between higher and lower  $NO_x$  conditions (for example, by leveraging the so-called weekend effect), comparing consecutive decades individually and ascribing their differences to only one factor ( $NO_x$  emissions) strikes me as flawed. The authors' observation that "decreasing  $NO_x$  concentrations are not the only factor causing the observed changes" underscores this fact, and raises the question of why they chose to dissect their long term data set in this fashion at all. I would recommend rethinking the approach here, and identifying a methodology that is less subject to non-stationarities in external variables.

As stated in the introduction, we would like to emphasize that  $O_3$ -temperature relationship varies in space and time, depending not only on the chemical but also meteorological conditions. However, the primary objective of our study is to assess how  $NO_x$  reductions influenced changes in the  $O_3$  sensitive to temperature. Therefore, we specifically focus on the role of  $NO_x$  reductions. For that, we have divided the complete period of study (1999-2018) into two sub-periods of 10 years, which allow us to build GAMs to assess the non-linear interaction  $NO_x$ -temperature in each period. As shown in previous studies (e.g. Pusede et al 2012, Jin et al. 2017, Solberg et al. 2017, Phalitonnkiat et al. 2016) the strategy of comparing different periods provides useful insights into the impacts of  $NO_x$  reductions in the temperature- $O_3$  relationship. Thus, we believe that the approach presented in this study is solid.

Nevertheless, we have taken note of this comment and we have further analyzed the weekend-weekday effect to support our results. As Referee #2 points out, the so-called "weekend-weekday" effect can be used as a marker of the dominant chemistry regime (Steiner et al. 2010; Murphy et al. 2007). Comparing changes in weekday and weekend  $O_3$  concentrations can provide an indication of the local chemical regimes of  $O_3$ . Under a NOx-saturated regime,  $O_3$  concentrations tend to increase during the weekends as a result of lower NOx (Pusede and Cohen 2012). Figure R3 shows the station type area annual averages of the "weekend-weekday" effect of daily maximum 8h average (MDA8) (i.e. difference between MDA8 concentrations during the weekends and  $O_3$  concentrations on weekdays). The weekend-weekday effect is more pronounced at the urban stations that show positive and larger values of  $\Delta$ MDA8 over most of the years. We observe a transition between chemical regimes (NOx-saturated, NOx-limited) in some years at rural and suburban stations. The weekend-weekday effect is more pronounced at the urban stations, consistent with a NOx-saturated regime. However, it is noted a general decrease of the "weekend-weekday" effect during the last years of the period of study, pointing out a general transition to NOx-limited regime.

Furthermore, to provide a general picture of the dominant regimes across the stations considered in our study, we have examined the sign and the magnitude of the weekend effect (i.e. difference between  $O_3$  concentrations during the weekends and  $O_3$  concentrations on weekdays) separately for each period (i.e. 1999-2008, 2009-2018) (Fig. R4). During the first period of the study 1999-2008 (left) most of the stations exhibit positive values of  $\Delta$ MDA8 and the urban stations show the largest values, consistent with a NOx-saturated regime. The weekend-weekday effect is lower across the rural stations, although we found positive values at some the rural stations over the southwest regions, which indicates a more dominant NOx-saturated regime. On the contrary, for the second period 2009-2018 a weaker weekend-weekday effect (right) is found. The rural stations show the lowest values, consistent with a NOx-limited chemistry. Moreover, the urban stations show a general tendency to move from a NOx-saturated regime towards a NOx-limited regime.

---

## Author Comment (AC2) · 3 Nov 2020

**Supplement of Observed changes in the temperature dependence response of surface ozone under NOx reductions over Germany**

Noelia Otero, Henning W. Rust and Tim Butler

**Modeling ozone production rates with GAMs**

Generalized Additve Models (GAMs)(Hastie and Tibshinari 1990; Wood 2006) are useful tools to examine complex non-linear relationships and have been previously applied to model air pollutants (Barmpadimos et al. 2011; Boleti, Hueglin, and Takahama 2019; Carslaw, Beevers, and Tate 2007; Jackson et al. 2009). We have used GAMs to model O3 production rates ( $\Delta$ O3) as a function of key variables that influence O3 production. GAMs are extensions of the generalized linear model (McCullagh and Nelder 1989) that work under the assumption that there is an additive effect between the response variable and the explanatory variables (covariates). Generalized linear models allow for response distributions other than the Normal distribution, and for a degree of non-linearity in the model structure (McCullagh and Nelder 1989). The basic form of GLM is represented as:

$$g(\mu_i) = X_i \beta \tag{1}$$

where  $\mu_i = E[Y_i]$ , g is a monotic function,  $X_i$  is the ith row of X (model matrix) and  $\beta$  is a vector of unknown parameters. GLM assumes that  $Y_i$  are independent and  $Y_i \sim$  some exponential family distribution (for more details see McCullagh and Nelder 1989; Wood 2006).

As stated in the manuscript (see section 3.3), GAM establishes a relationship between the response and a sum of smooth functions of the covariates through a link function (Hastie and Tibshinari 1990; Wood 2006). Thin plate regression splines were used as smoothers to describe a nonlinear relationship between the response and the covariates (Wood 2006). In addition, GAMs allow to model interactions created between covariates with different smoothers (or degrees of smoothness) assumed for each covariate (Wood 2006; Pedersen et al. 2019). Here, we introduced interactions terms using tensor products to represent the the interacting effects of two covariates (e.g. temperature-NOx) on the response ( $\Delta O_3$ ). For a general overview of GAM we refer to Hastie and Tibshinari (1990) and Wood (2006).

All calculations were carried out using the statistical software R (R Development Core Team 2018) with the mgcv package (Wood 2011).

**Model selection**

A set of covariates were used to build the GAMs: temperature, NOx, VPD, O3 concentrations from the previous hour (CO3(t - 1)), boundary layer height growth rate ( $\Delta$ BLH) and the MDA8 concentrations from the previous day (CMDA8(t - 24)). A forward selection process was used to select the covariates that better explain the  $\Delta$ O3. During the selection procedure, the interactions between two influencing covariates are also included in order to represent physical processes such as dry deposition, represented by the interaction between VPD and CO3(t - 1), and mixing processes captured by the interaction term between  $\Delta$ BLH and CMDA8(t - 24).

The selection process can be summarised as follows:

1. We first start with a baseline model that included the nonlinear relationship between  $NO_x$  and temperature as follows:

$$\Delta O_3 = f(T, NO_x) \tag{2}$$

where  $f(T, NO_x)$  represents the interaction between temperature (T) and  $NO_x$  concentrations and it is included as a tensor product (Wood 2017). Observing the skewness of the  $NO_x$  data led us to introduce a modification in the baseline model using a log transformation of  $NO_x$ .

- 2. We successively add further covariates and/or interactions that can improve the model performance.
- 3. The deviance explained and the Akaike information criterion (AIC) (Akaike 1974) are calculated in each step.
- 4. The GAM with the lowest AIC is selected as the best model.

We applied this procedure separately for each station and period, namely GAM-P1 for the first period (1999-2008) and GAM-P2 for the second period (2009-2019). Our goal with this process is to define a common model well defined across all of the stations (i.e. same structure in terms of covariates). Figure S1 shows the models built at each step (i.e. adding the covariates and interactions) during the selection process for the urban station in Berlin during the first period 1999-2008. It can be observed that the model performance considerably improves when adding the covariates and the complexity (i.e. more interaction terms).

Figure S1. AIC and deviance explained (DEV) for each model used during the stepwise process at one rural station during the first period 1999-2008.

We obtained similar results for most of stations, which led us to select the best model with the following structure that includes three interaction terms:

$$\Delta O_3 = T * NO_x + VPD * C_{O_3}(t-1) + \Delta BLH * C_{MDA8}(t-24)$$

$$\tag{3}$$

The model performance was assessed through standard diagnostic plots: QQ plots of the deviance residuals, scatter plots of the residuals against the fitted values, histogram of residuals and scatter plots the response against the fitted values (Wood 2006). In general the diagnostic plot did not show concerning patterns in the residuals. As an example, Fig. S2 shows the standard plots to check the model assumptions obtained by the function gam.check()(Wood 2011).

Figure S2. Diagnostic plots for the Berlin urban station (DEBE034) for the period 1999-2008: QQ-plot of residuals, linear predictor vs. residuals, the histogram of residuals and the plot of fitted values vs. response.

**List of figures**